# A General Rank Preserving Framework for Asymmetric Image Retrieval

**Hui Wu**[1]  **Min Wang**[2*]  **Wengang Zhou**[1,2*]  **Houqiang Li**[1,2]
[1]CAS Key Laboratory of Technology in GIPAS, University of Science and Technology of China
[2]Institute of Artificial Intelligence, Hefei Comprehensive National Science Center
`wh241300@mail.ustc.edu.cn, wangmin@iai.ustc.edu.cn, {zhwg,lihq}@ustc.edu.cn`

## Abstract

Asymmetric image retrieval aims to deploy compatible models on platforms of different resources to achieve a balance between computational efficiency and retrieval accuracy. The most critical issue is how to align the output features of different models. Despite the great progress, existing approaches apply strong constraints so that features or neighbor structures are strictly aligned across different models. However, such a one-to-one constraint is too strict to be well preserved for the query models with low capacity. Considering that the primary concern of the users is the rank of the returned images, we propose a generic rank preserving framework, which achieves feature compatibility and the order consistency between query and gallery models simultaneously. Specifically, we propose two alternatives to instantiate the framework. One realizes straightforward rank order preservation by directly preserving the consistency of the sorting results. To make sorting process differentiable, the Heaviside step function in sorting is approximated by the sigmoid function. The other aims to preserve a learnable monotonic mapping relationship between the returned similarity scores of query and gallery models. The mapped similarity scores of gallery model are considered as pseudo-supervision to guide the query model training. Extensive experiments on various large-scale datasets demonstrate the superiority of our two proposed methods.

## 1 Introduction

In recent years, deep representation learning methods (Babenko et al., 2014; Tolias et al., 2016; 2020) have achieved great progress in image retrieval. Typically, most existing image retrieval tasks belong to **symmetric image retrieval**, in which a deep representation model is deployed to map both query and gallery images into the same discriminative feature space. During online retrieval, gallery images are ranked by sorting their feature distances, *e.g.*, cosine similarity or Euclidean distance, against query image. To achieve high retrieval accuracy, most existing methods tend to deploy a large powerful representation model. In a real-world visual search system, the gallery side is usually on the cloud-based platforms, which have sufficient resources to deploy large powerful models. As for the query side, *e.g.*, mobile phone or smart camera, its resources are too constrained to meet the demand of deploying large models. To strike a balance between performance and efficiency, it is better to deploy a lightweight model on the query side, while a large one for the gallery side. This setup is denoted as **asymmetric image retrieval** (Duggal et al., 2021; Budnik & Avrithis, 2021).

For asymmetric retrieval, how to align the embedding spaces of the query and gallery models is the core problem. To this end, BCT (Shen et al., 2020) first introduces feature compatibility learning. Concurrent work AML (Budnik & Avrithis, 2021) learns the query model by contrastive learning with gallery model extracting features for positive and negative samples. Recently, CSD (Wu et al., 2022b) achieves promising results by considering both first-order feature imitation and second-order neighbor similarity preservation during the learning of the query model. Despite the great progress, existing methods enforce the consistency of features or neighbor structures across different models, which is too strict to be well preserved for lightweight query models with low capacity. For users, the order of the returned images plays a more important role than the image features or similarity

---

[*]Corresponding authors: Min Wang and Wengang Zhou

scores. Strictly enforcing the feature-level one-to-one consistency is not the best choice to achieve better asymmetric retrieval accuracy, while rank preserving deserves more attention.

To address the above issues, we propose a general rank preserving framework, which directly optimizes the consistency of rank order to realize the feature compatibility across query and gallery models implicitly. Specifically, for a training image, we first extract its features with the query and gallery models, respectively. Then, the gallery feature is utilized for symmetric retrieval in a database, in which images are also embedded by the gallery model. After that, the ranking list and similarity scores are returned. We select the top $K$ images in the ranking list and calculate their asymmetric similarity scores with the query feature. These asymmetric similarity scores may result in different rank orders from those returned by symmetric retrieval. Thus, two instantiation methods are proposed to achieve the consistency of these two rank orders.

The first aims to directly optimize the rank order consistency of the sorting results. To make the sorting process differentiable, the sigmoid function is adopted to approximate the Heaviside step function (Davies, 1978), which is typically used for numerical comparison. Secondly, we propose to maintain a learnable monotonic mapping relationship between the symmetric and asymmetric similarity scores. A learnable monotonically increasing function is applied to the similarity scores returned by the symmetric retrieval, which will serve as the pseudo-supervision of the query model. Then, we constrain the consistency between the mapped similarity scores and the asymmetric similarity scores to optimize the query model. Notably, both two instantiations preserve the rank order of the images, which are returned by symmetric retrieval.

Compared with previous methods, our framework has a unique advantage. It does not constrain the query model to mimic the features or the overall neighbor structures of the gallery model. Instead, it expects that the query model maintains the order of the returned images from symmetric retrieval. Thus, our framework weakens the restriction on the query model with low capacity, and reduces the risk of overfitting during learning the query model. Besides, our framework utilizes no annotation or label of the training data, which makes it flexible and adaptable in various real-world scenarios. To evaluate our approach, comprehensive experiments are conducted on four popular retrieval datasets. Ablations demonstrate the effectiveness and generalizability of our framework. Our approach surpasses the existing state-of-the-art methods by a considerable margin.

## 2 RELATED WORK

**Image Retrieval**. Given a large corpus, image retrieval aims to efficiently find the images, which contain the same object or describe the same content with the queries, based on their feature similarities. Most of the traditional image retrieval systems are based on local features (Lowe, 2004; Bay et al., 2006) and bag-of-words (Sivic & Zisserman, 2003; Philbin et al., 2007) representations borrowed from text retrieval. There are also several aggregation methods including VLAD (Jégou et al., 2011), Fisher vectors (Perronnin et al., 2010) and ASMK (Tolias et al., 2013), which are used to aggregate local features into compact representations for efficient search. Recently, with the proposed various pooling methods (Kalantidis et al., 2016; Tolias et al., 2016; Radenović et al., 2018b) and loss functions (Revaud et al., 2019; Deng et al., 2019; Weinzaepfel et al., 2022), deep learning has greatly improved the performance of image retrieval. Despite the great progress, a large deep model is usually deployed for its optimal performance, which, however, is not applicable in some resource-constrained scenarios. In this work, we focus on asymmetric retrieval, where the query (user) side deploys a lightweight model while the gallery side applies a large model.

**Feature Compatibility**. The core of asymmetric retrieval is to align the features of the query and gallery models, which is also known as feature compatibility. BCT (Shen et al., 2020) first formulates the problem of feature compatible learning and reuses the old classifier for the query model training. AML (Budnik & Avrithis, 2021) achieves the feature compatibility by performing asymmetric contrastive learning between different models. After that, CSD (Wu et al., 2022b) achieves the preservation of neighbor similarities in the embedding space of the gallery model in an unsupervised manner. As for HVS (Duggal et al., 2021), both parameter and architecture are considered simultaneously in a unified framework, which gives promising performance. Other lines of research follow the model regression problem (Yan et al., 2021; Zhang et al., 2022; Duggal et al., 2022) during gallery model updating, which is also related to feature compatibility. In this work, we start from

the main concern of the users, *i.e.*, the order of the returned images in the ranking list, and propose a general rank preserving framework, which is free of annotations from training datasets.

**Lightweight Network**. Thanks to the evolution of the network architecture (He et al., 2016; Tan & Le, 2021), deep convolutional neural networks (CNNs) achieve superior performance in various computer vision tasks. Usually, a large powerful model leads to better performance with the consumption of more storage and computational resources. Real-world tasks aim to achieve the best accuracy with a limited computational budget, which is determined by the target platforms. The demand of deploying high-performance deep models on resource-constrained platforms has led to a series of studies on model compression (Antonio et al., 2016; He et al., 2018b; Oktay et al., 2020) and efficient network architecture design, *e.g.*, MobileNets (Howard et al., 2017; Sandler et al., 2018), ShuffleNets (Zhang et al., 2018; Ma et al., 2018) and EfficientNets (Tan & Le, 2019). In this work, we focus on asymmetric retrieval in resource-constrained scenario. Since query features are extracted on resource-constrained end platforms, our approach employs the various lightweight models mentioned above as query models.

**Smooth Rank Approximation**. There is a long history of designing smooth surrogate for rank approximation. In image retrieval, the well-known surrogate is to constrain the relative relationships between pairs (Raia et al., 2006) or triplets (Gordo et al., 2017), which implicitly leads to partial ranking. Some methods propose to utilize a smooth discretization of similarity scores (He et al., 2018a; Cakir et al., 2019; Ustinova & Lempitsky, 2016; Revaud et al., 2019) for the rank approximation. Other approaches explicitly approximate the non-differentiable rank metric with a neural network (Engilberge et al., 2019) or a sum of sigmoid functions (Brown et al., 2020; Huang et al., 2022; Patel et al., 2022). Recently, a more accurate and robust approximation method is proposed in ROADMAP (Elias et al., 2021). However, all the methods mentioned above are designed for symmetric retrieval, where only a single model exists and no cross-model feature compatibility has been considered. Thus, they cannot be directly applied for asymmetric retrieval. In our approach, asymmetric and symmetric retrieval are performed with the same query. Then, the order of two returned ranking lists are constrained to be consistent, which also ensures feature compatibility.

## 3 Background: Asymmetric Image Retrieval

Given images of interest (referred to as query set $\mathcal{Q}$), image retrieval targets at correctly finding images of the same content or object from a large-scale gallery set $\mathcal{G}$. An image encoder $\phi(\cdot)$ is deployed to map the images into $L_2$-normalized feature vectors. Then, the cosine similarity of two normalized vectors is used for measuring the similarity between query and gallery images. Usually, some metrics, *e.g.*, mean Average Precision (mAP), are adopted to evaluate a retrieval system, which are conditioned on $\phi(\cdot)$, $\mathcal{Q}$ and $\mathcal{G}$. For convenience, we ignore query and gallery sets and denote the metric as $\mathcal{M}(\phi_q(\cdot), \phi_g(\cdot))$, where $\phi_q(\cdot)$ and $\phi_g(\cdot)$ are the image encoders deployed for query and gallery feature extraction, respectively.

In a conventional symmetric retrieval system, the same encoder, *i.e.*, $\phi_g(\cdot) = \phi_q(\cdot)$, is used for both gallery and query sides. Typically, deploying a powerful model results in better retrieval accuracy. However, it is not applicable in some resource-constrained platforms, *e.g.*, mobile devices or smart cameras. Assuming $\phi_q(\cdot)$ is different from and significantly smaller than $\phi_g(\cdot)$, **asymmetric image retrieval** leverages $\phi_q(\cdot)$ to embed query images and $\phi_g(\cdot)$ to embed the gallery images. Thus, we should ensure that the lightweight query model $\phi_q(\cdot)$ maps images into the same embedding space of the large gallery model $\phi_g(\cdot)$. Besides, an asymmetric retrieval system is expected to achieve a retrieval accuracy comparable to that of a symmetric retrieval system (Duggal et al., 2021),

$$\mathcal{M}(\phi_g(\cdot), \phi_g(\cdot)) \approx \mathcal{M}(\phi_q(\cdot), \phi_g(\cdot)) > \mathcal{M}(\phi_q(\cdot), \phi_q(\cdot)). \tag{1}$$

## 4 Rank Persevering Framework

An overview of our framework is shown in Fig. 1. Given a well-trained gallery model $\phi_g(\cdot)$, we aim to learn a lightweight query model $\phi_q(\cdot)$ to be compatible with it. Assume there exits a training gallery set $\mathcal{G}_t$, we first embed it into $\boldsymbol{F} = [\boldsymbol{f}_g^1, \boldsymbol{f}_g^2, \ldots, \boldsymbol{f}_g^N] \in \mathbb{R}^{N \times d}$ with $\phi_g(\cdot)$:

$$\boldsymbol{f}_g^i = \phi_g(g_i) \in \mathbb{R}^d, \ i = 1, 2, \ldots, N, \tag{2}$$

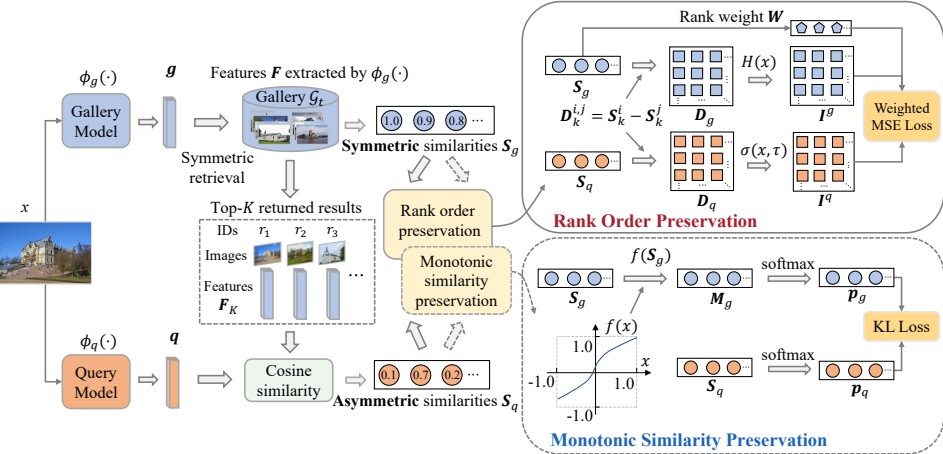

Figure 1: **An overview of Rank preserving framework.** Given a training image $x$, gallery model $\phi_g(\cdot)$ and query model $\phi_q(\cdot)$ encode it into features $g$ and $q$, respectively. $g$ is treated as the query to search in a training gallery set $\mathcal{G}_t$, of which the images are also embedded by $\phi_g(\cdot)$. We fetch the features $F_K$ of the top-$K$ images in the ranking list and calculate the asymmetric similarity scores $S_q$ with $q$. Two instantiations of our framework named **Ranking Order Preservation** (Sec. 4.1) and **Monotonic Similarity Preservation** (Sec. 4.2) are proposed to ensure the consistency of rank orders when the query is embedded by $\phi_q(\cdot)$ and $\phi_g(\cdot)$, respectively.

where $g_i$ is the $i$-th image in the training gallery set $\mathcal{G}_t$. During the learning of $\phi_q(\cdot)$, gallery model is fixed. For each training image $x$, we extract its features $q$ and $g$ with $\phi_q(\cdot)$ and $\phi_g(\cdot)$, respectively:

$$q = \phi_q(x) \in \mathbb{R}^d, \ g = \phi_g(x) \in \mathbb{R}^d. \tag{3}$$

Then, we perform symmetric retrieval in $\mathcal{G}_t$ with $g$ as the query. After that, we obtain the ranking list $R = [r_1, r_2, \ldots, r_K] \in \mathbb{R}^K$ and similarity scores $S_g = [g^T f_g^{r_1}, g^T f_g^{r_2}, \ldots, g^T f_g^{r_K}] \in \mathbb{R}^K$ of top-$K$ images, where $r_i$ denotes the ID of the $i$-th image in $\mathcal{G}_t$. Notably, the values in $S_g$ satisfy the monotonically decreasing property, *i.e.*, $g^T f_g^{r_1} \geqslant g^T f_g^{r_2} \geqslant \cdots \geqslant g^T f_g^{r_K}$. With the ranking list $R$, the corresponding feature embeddings $F_K = [f_g^{r_1}, f_g^{r_2}, \ldots, f_g^{r_K}]$ for the top-$K$ images are taken from $F$. Then, we calculate the asymmetric similarity scores between the query feature $q$ and $F_K$:

$$S_q = [q^T f_g^{r_1}, q^T f_g^{r_2}, \ldots, q^T f_g^{r_K}] \in \mathbb{R}^K. \tag{4}$$

In CSD (Wu et al., 2022b), the consistency of $S_q$ and $S_g$ is directly constrained to optimize the query model. However, we argue that it is too strict to be preserved well for the lightweight model with low capacity. Besides, the user experience is mainly influenced by the rank order of the returned images rather than the specific similarity scores. Thus, it is better to directly impose the constraint on the rank order of the returned images rather than the specific similarity scores. Specifically, we need to ensure the values in $S_q$ are also monotonically decreasing:

$$q^T f_g^{r_1} \geqslant q^T f_g^{r_2} \geqslant \cdots \geqslant q^T f_g^{r_K}. \tag{5}$$

To this end, we propose two methods to instantiate the rank preserving framework. One aims to achieve straightforward **Rank Order Preservation (ROP)** by constraining the consistency of the sorting results, which are formulated as the indicator matrices in Sec. 4.1. Typically, it is not feasible to optimize the sorting results directly due to the non-differentiable Heaviside step function (Davies, 1978), which is used for numerical comparison in sorting. Thus, sigmoid function is introduced as the smooth approximation, which enables the optimization of sorting process with back-propagation methods. The other is **Monotonic Similarity Preservation (MSP)**, which aims to preserve a learnable monotonic mapping relationship between the returned similarity scores $S_q$ and $S_g$. Specifically, we learn a monotonically increasing function, which predicts $S_q$ given $S_g$. Then, the consistency between the mapped $S_g$ and $S_q$ is restricted to train the query model.

## 4.1 RANK ORDER PRESERVATION

The motivation of our method is directly preserving the rank order of the returned images between asymmetric and symmetric retrieval. Thus, the critical problem is to select a suitable evaluation

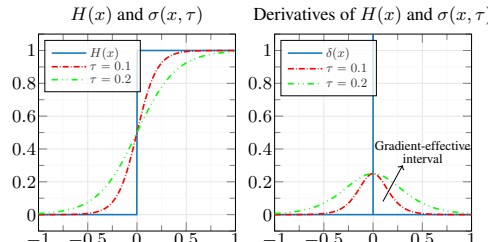
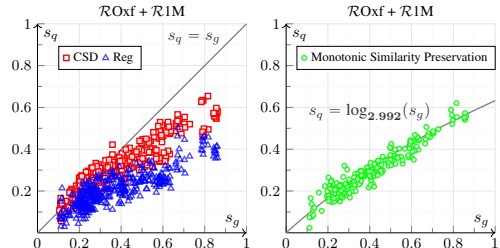

Figure 2: (*Left*) Heaviside step function and two sigmoid functions with different tempurature $\tau$ as approximations. (*Right*) Corresponding derivatives.

Figure 3: Visualization of the similarity score distribution (symmetric *vs.* asymmetric). $s_g$: symmetric similarity score; $s_q$: asymmetric similarity score.

metric as the constraint. In this section, we dive into the sorting process and constrain the sorting results of symmetric and asymmetric retrieval to be consistent.

**Sorting process**. Sorting the similarity scores between the query and database images is an essential operation to get the final ranking list in image retrieval. Numerical comparison is the most fundamental operation in various sorting algorithms. In this work, we take the comparison sorting as the sorting algorithm. Assume there exists a list of similarity scores $\boldsymbol{S} = [s_1, s_2, \ldots, s_N] \in \mathbb{R}^N$ needed to be sorted. We first compute the difference matrix $\boldsymbol{D} \in \mathbb{R}^{N \times N}$ between any elements in $\boldsymbol{S}$. Then, the binary indicator matrix $\boldsymbol{I} \in \mathbb{R}^{N \times N}$ is calculated by applying the Heaviside step function $H(x)$, which is equal to 0 for negative values, otherwise equal to 1, on the difference matrix $\boldsymbol{D}$:

$$\boldsymbol{D} = \begin{bmatrix} s_1 & \ldots & s_N \\ \vdots & \ddots & \vdots \\ s_1 & \ldots & s_N \end{bmatrix} - \begin{bmatrix} s_1 & \ldots & s_1 \\ \vdots & \ddots & \vdots \\ s_N & \ldots & s_N \end{bmatrix}, \quad \boldsymbol{I} = \begin{bmatrix} H(s_1 - s_1) & \ldots & H(s_N - s_1) \\ \vdots & \ddots & \vdots \\ H(s_1 - s_N) & \ldots & H(s_N - s_N) \end{bmatrix}. \quad (6)$$

Each element $\boldsymbol{I}_{i,j}$ in the indicator matrix $\boldsymbol{I}$ denotes the relative relationship between the similarity scores $s_i$ and $s_j$. $\boldsymbol{I}_{i,j} = 1$, if $s_j$ is larger than or equal to $s_i$, otherwise, $\boldsymbol{I}_{i,j} = 0$.

**Indicator matrix consistency**. Since the indicator matrix demonstrates the relative relationships of similarity scores, we take it as the constraint for optimizing the query model. If the indicator matrix is well preserved, the asymmetric retrieval will result in the same ranking list as the symmetric retrieval. Specifically, we take $\boldsymbol{S}_g$ and $\boldsymbol{S}_q$ into Eq. (6) to get the corresponding indicator matrices $\boldsymbol{I}^g \in \mathbb{R}^{K \times K}$ and $\boldsymbol{I}^q \in \mathbb{R}^{K \times K}$:

$$\boldsymbol{I}^g = \begin{bmatrix} 1 & \ldots & H(\Delta_g^{K,1}) \\ \vdots & \ddots & \vdots \\ H(\Delta_g^{1,K}) & \ldots & 1 \end{bmatrix} = \begin{bmatrix} 1 & \ldots & 0 \\ \vdots & \ddots & \vdots \\ 1 & \ldots & 1 \end{bmatrix}, \quad \boldsymbol{I}^q = \begin{bmatrix} 1 & \ldots & H(\Delta_q^{K,1}) \\ \vdots & \ddots & \vdots \\ H(\Delta_q^{1,K}) & \ldots & 1 \end{bmatrix}, \quad (7)$$

where $\Delta_l^{m,n} = \boldsymbol{S}_l^m - \boldsymbol{S}_l^n$, $l \in \{q, g\}$. Generally, if one gallery image has a higher similarity score against query image, it is more possible to be the true positive, which deserves more attention during the training of query model. To this end, we design a ranking weight $\boldsymbol{W}_i = \exp(\boldsymbol{S}_g^i/\tau_r)/(i \times \sum_{l=1}^{K} \exp(\boldsymbol{S}_g^l/\tau_r))$, $i = 0, 1, \ldots, K$, where $\tau_r$ is the temperature. Finally, the weighted indicator matrix consistency loss is minimized as the final objective function to train the query model:

$$\mathcal{L}_{\text{ROP}} = \sum_{i=1}^{K} \boldsymbol{W}_i \left\| \boldsymbol{I}_{i,:}^g - \boldsymbol{I}_{i,:}^q \right\|_2^2 = \sum_{i=1}^{K} \sum_{j=1}^{K} \boldsymbol{W}_i (H(\Delta_g^{i,j}) - H(\Delta_q^{i,j}))^2. \quad (8)$$

**Heaviside step function approximation**. Unfortunately, the derivative of the Heaviside step function $H(x)$ is Dirac delta function $\delta(x)$, which is either flat everywhere, with zero gradient, or discontinuous, and hence cannot be optimized with gradient-based method. Inspired by (Brown et al., 2020), the sigmoid function $\sigma(x, \tau) = \frac{1}{1+e^{-\frac{x}{\tau}}}$, where $\tau$ denotes the temperature adjusting the sharpness, is used to approximate the Heaviside step function smoothly. As shown in Fig. 2, the temperature governs the approximation tightness and the gradient-effective interval. Substituting $\sigma(x, \tau)$ into Eq. (8), the weighted indicator matrix consistency loss is approximated as:

$$\mathcal{L}_{\text{ROP}} = \sum_{i=1}^{K} \sum_{j=1}^{K} \boldsymbol{W}_i (H(\Delta_g^{i,j}) - \sigma(\Delta_q^{i,j}, \tau))^2. \quad (9)$$

## 4.2 Monotonic Similarity Preservation

In this section, we introduce another method to preserve the rank order. In Fig. 3, we visualize the distributions of the similarity score pairs $s_q$ and $s_g$ in existing methods. It is observed that same $s_g$ may correspond to a wide range of values for $s_q$. In other words, the similarity score pairs locate on wide strips, which means that existing methods do not well preserve the order of the images returned by symmetric retrieval. We think it is due to the fact that existing methods all impose a strict one-to-one constraint, which may cause overfitting for the query model with low capacity.

Therefore, we introduce a learnable monotonic mapping function $f(x)$ applied to $S_g$ to form the pseudo-supervision of $S_q$. Formally, $M_g^i = f(S_g^i)$, $i = 1, 2, \ldots, K$. This avoids the strict neighbor structure alignment between query and gallery models. After that, the Kullback-Leibler (KL) divergence, which shows excellent performance in CSD (Wu et al., 2022b), between $M_g$ and $S_q$ is adopted as the final objective function to optimize the query model. Specifically, we first convert $M_g$ and $S_q$ into the form of probability distributions:

$$\boldsymbol{p}_g^i = \frac{\exp\left(\boldsymbol{M}_g^i/\tau_g\right)}{\sum_{l=1}^K \exp\left(\boldsymbol{M}_g^l/\tau_g\right)}, \ \boldsymbol{p}_q^i = \frac{\exp\left(\boldsymbol{S}_q^i/\tau_q\right)}{\sum_{l=1}^K \exp\left(\boldsymbol{S}_q^l/\tau_q\right)}, \ i = 1, 2, \ldots, K, \tag{10}$$

where $\tau_q$ and $\tau_g$ are temperature coefficients. Both $\tau_q$ and $\tau_g$ are set less than 1 to keep $\phi_q(\cdot)$ focus on the top images of the ranking list. Then, the monotonic similarity preservation loss is defined as

$$\mathcal{L}_{\text{MSP}} = \text{KL}(\boldsymbol{p}_g || \boldsymbol{p}_q) = \sum_{i=1}^K \boldsymbol{p}_g^i \log(\frac{\boldsymbol{p}_g^i}{\boldsymbol{p}_q^i}). \tag{11}$$

To preserve the order of each elements in $S_g$, we should ensure that the function $f(x)$ is monotonically increasing. In this work, we consider three common families of the monotonically increasing functions, which are discussed in the following.

**Logarithmic function**. Considering that the definition domain of the logarithmic functions is $(0, +\infty)$ and the cosine similarity lies between -1 and 1, we define the mapping function as $f(x) = \log_a(x + 1)$, $(-1.0 < x < 1.0, \ a > 1)$, where $a$ is a learnable parameter.

**Exponential function**. Another common monotonically increasing function is the exponential function. To make each $M_g^i$ exist in the range less than 1.0, $f(x)$ is defined as $f(x) = a^{x-1}$, $(-1.0 < x < 1.0, \ a > 1)$, where $a$ is also a learnable parameter.

**Polynomial function**. There are a wide variety of polynomial combinations. In this work, we consider a simple case. We first choose a set of basis functions $\mathcal{X} = \{x^\alpha, x^{2\alpha}, \ldots, x^{N\alpha}\}$, then the mapping function is defined as linear combinations of those basis functions. Formally, $f(x) = \sum_{i=1}^N a_i x^{i\alpha}$, $(-1.0 < x < 1.0, \ \alpha > 0)$, where $\{a_1, a_2, \ldots, a_N\}$ is a set of learnable parameters. To ensure that $f(x)$ is monotonically increasing, each $a_i$ should be greater than 0. Besides, we also make the sum of $a_i$ equal to 1 to control the range of $f(x)$.

**Relation with contextual similarity distillation**. In CSD (Wu et al., 2022b), KL loss strictly restricts the consistency between $S_g$ and $S_q$. When the loss minimum is achieved, a linear relationship $\tau_q S_g = \tau_g S_q$ between $S_g$ and $S_q$ is presented. In MSP, when the mapping function is set to $f(x) = x$ (diagonal line in Fig. 3), $M_g$ is equal to $S_g$, the loss function $\mathcal{L}_{\text{MSP}}$ degrades to the form in CSD. Thus, CSD is a special case of our monotonic similarity preservation. As shown in Fig. 3, our MSP method realizes a narrower striped similarity distribution, since it introduces a learnable monotonically increasing mapping between symmetric and asymmetric similarities. As a result, our MSP method could preserve the order of the returned images better than CSD.

## 5 Experiments

### 5.1 Implementation Details

**Training datasets**. Two datasets are used for training. One is SfM-120k (Radenović et al., 2018b), of which 551 3D models are taken for training while the other 162 3D models for validation. The other is GLDv2 (Weyand et al., 2020), which consists of $1,580,470$ images with $81,311$ classes. We randomly sample $80\%$ images from GLDv2 for training and leave the rest $20\%$ for validation.

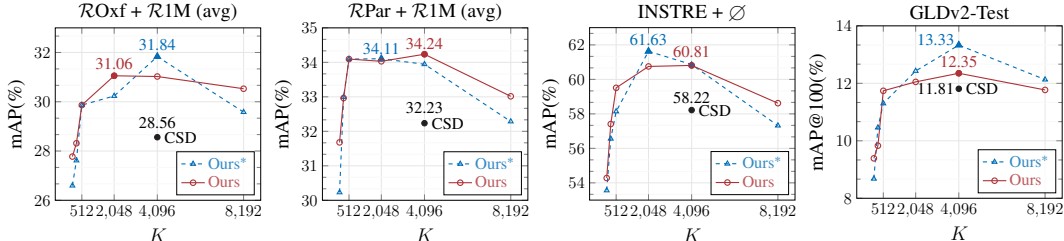

Figure 4: Analysis of the **length of the ranking list** $K$ in our methods. SfM-120k, GeM and MobileNetV2 are adopted as training set, gallery and query models, respectively.

| $\tau$ | SfM-120k val | INSTRE $+ \varnothing$ | $\mathcal{R}$Oxf $+ \mathcal{R}$1M Med | Hard | $\mathcal{R}$Par $+ \mathcal{R}$1M Med | Hard |
|---|---|---|---|---|---|---|
| 0.001 | 67.33 | 34.13 | 31.59 | 12.46 | 37.11 | 12.23 |
| 0.01 | 71.16 | 41.30 | 34.87 | 13.96 | 40.47 | 14.56 |
| 0.05 | 76.33 | 59.59 | 42.08 | 19.28 | 46.63 | 19.36 |
| 0.1 | **77.06** | **60.75** | **42.33** | **19.78** | **47.72** | **20.34** |
| 0.5 | 75.46 | 57.19 | 41.68 | 18.83 | 45.64 | 18.64 |
| 1.0 | 74.86 | 55.78 | 40.94 | 18.15 | 44.36 | 16.99 |

Table 1: Analysis of **temperature** $\tau$ in Eq. (9). $\tau$ controls the smoothness of the sigmoid function used to approximate the Heaviside step function.

| $W_i$ | SfM-120k val | INSTRE $+ \varnothing$ | $\mathcal{R}$Oxf $+ \mathcal{R}$1M Med | Hard | $\mathcal{R}$Par $+ \mathcal{R}$1M Med | Hard |
|---|---|---|---|---|---|---|
| 1 | 50.99 | 25.12 | 32.73 | 14.81 | 34.00 | 16.85 |
| $1/i$ | 75.99 | 52.57 | 42.52 | 20.17 | 44.76 | 18.03 |
| $\boldsymbol{S}_g^i$ | 59.43 | 59.59 | 36.55 | 16.27 | 36.27 | 16.11 |
| $P_i$ | 76.91 | 58.63 | **42.73** | **20.07** | **47.92** | 20.11 |
| $P_i/i$ | **77.06** | **60.75** | 42.33 | 19.78 | 47.72 | **20.34** |
| $1 - i/K$ | 69.57 | 31.08 | 34.95 | 15.89 | 44.36 | 16.99 |

Table 2: Analysis of **ranking weight** $W_i$ in Eq. (9). $K$: length of the ranking list; $P_i$ denotes $\exp(\boldsymbol{S}_g^i/\tau_r)/\sum_{l=1}^{K}\exp(\boldsymbol{S}_g^l/\tau_r)$.

**Evaluation datasets and metrics**. We evaluate the trained query models on four datasets under the setting of asymmetric retrieval, including GLDv2-Test (Weyand et al., 2020), Revisited Oxford with R1M ($\mathcal{R}$Oxf $+ \mathcal{R}$1M), Revisited Paris with R1M ($\mathcal{R}$Par $+ \mathcal{R}$1M) (Radenović et al., 2018a) and INSTRE (Wang & Jiang, 2015). The evaluation metric for GLDv2-Test is mAP@100, while all other datasets are mAP. See App. A.1 for the more detailed descriptions of the testing sets.

**Architectures**. Following the settings in CSD (Wu et al., 2022b), we choose ResNet101 (He et al., 2016) trained by GeM (Radenović et al., 2018b) and DELG (Cao et al., 2020) as the gallery models. For the query model, common lightweight models, *e.g.*, ShuffleNets (Ma et al., 2018), MobileNets (Sandler et al., 2018) and EfficientNets (Tan & Le, 2019), are chosen. To adapt the lightweight model for the image retrieval task, we adjust the model architecture slightly. The details are present in App. A.1, with computation and parameter complexity statistics.

## 5.2 ANALYSIS AND ABLATIONS

In this section, we analyze the proposed rank preserving framework and perform exhaustive ablations. R101 and Mv2 denote ResNet101 and MobileNetV2, respectively. "**Ours**" and "**Ours\***" denote that we train lightweight query model $\phi_q(\cdot)$ with **ROP** and **MSP** constraints, respectively.

**Length $K$ of the ranking list.** In Fig. 4, we compare our proposed methods against CSD when ranking list has different length $K \in \{64, 256, 512, 2048, 4096, 8192\}$. As the length increases, the performance increases but saturates when $K = 4096$, after which, the performance decreases. When $K$ is small, the query model only needs to focus on the elements in front of the ranking list, without taking full advantage of the order information in the ranking list. Thus, the performance is unsatisfactory. On the contrary, when $K$ is particularly large, the query model is concerned with a very wide range of elements in the ranking list. Images at the bottom of the ranking list are far away from the query, their relative orders have almost no effect on the overall retrieval accuracy. Constraining the query model to preserve the order of this part leads to a decreased performance.

**Heaviside step function approximation**. In Tab. 1, we investigate the effect of the temperature $\tau$, which governs the smoothness of the sigmoid function. Results show that $\tau = 0.1$ leads to the best performance over different datasets. As explained in Sec. 4.1, a smaller value of $\tau$ leads to a narrower gradient-effective interval (Fig. 2) and a tighter approximation to the Heaviside step function. The strong acceleration of the gradient around zero encourages moving instances in the embedding space, leading to a change of rank. However, excessively small $\tau$ causes gradient vanishing, which is harmful to the optimization of the query model. In contrast, a large value of $\tau$ provides a wide gradient-effective interval at the cost of a looser approximation to the true order.

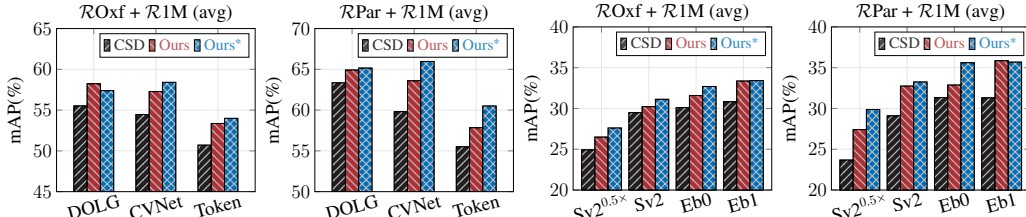

Figure 5: Analysis of **different model architectures**. *Two on the left:* Three typical deep representation models are chosen as gallery models. (1) DOLG (Yang et al., 2021) based on global and local feature aggregation. (2) CVNet (Lee et al., 2022) based on global pooling. (3) Token (Wu et al., 2022a) based on local feature aggregation. MobileNetV2 is adopted as the query model for all settings. *Two on the right:* GeM serves as the gallery model, and four lightweight models, including ShuffleNetv2 $(0.5\times)$ ($\text{Sv2}^{0.5\times}$), ShuffleNetv2 (Sv2), EfficientNetB0 (Eb0) and EfficientNetB1 (Eb1), are chosen as query models, respectively.

| Method | Training dataset $\mathcal{T}$ | GLDv2 -Test | INSTRE $+\varnothing$ | INSTRE $+\mathcal{D}1M$ | $\mathcal{R}$Oxf + $\mathcal{R}$1M Med | $\mathcal{R}$Oxf + $\mathcal{R}$1M Hard | $\mathcal{R}$Par + $\mathcal{R}$1M Med | $\mathcal{R}$Par + $\mathcal{R}$1M Hard |
|---|---|---|---|---|---|---|---|---|
| Reg | | 12.11 | 60.88 | 51.62 | 40.34 | 18.15 | 46.91 | 19.77 |
| CSD | GLDv2 | 14.55 | 64.51 | 55.77 | 46.02 | 21.85 | 52.05 | 24.13 |
| **Ours** | -index | 15.01 | 65.70 | 56.74 | 47.57 | 23.33 | 53.44 | 25.01 |
| **Ours*** | | **15.30** | 65.36 | 57.77 | 47.76 | 23.12 | 53.12 | 25.43 |
| Reg | | 11.22 | 69.34 | 59.44 | 38.76 | 16.72 | 44.24 | 17.29 |
| CSD | INSTRE$^\dagger$ | 13.00 | 71.86 | 62.83 | 44.95 | 21.09 | 49.07 | 21.94 |
| **Ours** | + $\mathcal{D}$1M | 14.27 | **75.30** | **64.50** | 46.75 | 22.74 | 51.59 | 23.87 |
| **Ours*** | | 14.06 | 73.89 | 64.18 | 46.08 | 22.12 | 51.10 | 23.13 |
| Reg | | 12.54 | 61.41 | 51.67 | 42.19 | 19.23 | 47.31 | 19.46 |
| CSD | $\mathcal{R}$Oxf$^\dagger$ | 14.19 | 65.65 | 57.81 | 46.81 | 21.70 | 53.06 | 25.88 |
| **Ours** | + $\mathcal{R}$Par$^\dagger$ | 14.85 | 67.39 | 60.78 | 49.83 | 23.97 | 54.46 | 27.29 |
| **Ours*** | + $\mathcal{R}$1M | 14.31 | 68.05 | 61.42 | **50.77** | **24.96** | **55.03** | **27.56** |

Table 4: Comparison of different **unsupervised methods trained on the deployed gallery set**. † denotes the gallery set of that testing dataset. MobileNetV2 and GeM are adopted as query and gallery models, respectively. Reg: (Budnik & Avrithis, 2021); CSD: (Wu et al., 2022b).

| Method | Query Net $\phi_q(\cdot)$ | Gallery Net $\phi_g(\cdot)$ | GLDv2 -Test | INSTRE $+\varnothing$ | INSTRE $+\mathcal{D}1M$ |
|---|---|---|---|---|---|
| GeM | R101 | R101 | 14.47 | 71.23 | 64.10 |
| Contr* | | | 9.44 | 28.05 | 15.93 |
| Reg | | | 10.58 | 34.96 | 23.06 |
| CSD | Mv2 | R101 | 11.81 | 58.22 | 46.84 |
| **Ours** | | | 12.35 | 60.81 | **50.96** |
| **Ours*** | | | **13.33** | 60.86 | 49.72 |
| DELG | R101 | R101 | 26.77 | 37.54 | 31.66 |
| Reg | | | 21.59 | 21.84 | 13.88 |
| HVS | | | 21.99 | 22.80 | 14.71 |
| LCE | Mv2 | R101 | 22.42 | 23.85 | 15.91 |
| CSD | | | 23.45 | 22.34 | 15.44 |
| **Ours** | | | 23.86 | 29.93 | 21.76 |
| **Ours*** | | | **25.44** | **34.34** | **26.58** |

Table 5: Comparison to the state-of-the-art methods on **INSTRE and GLDv2-Test**. Contr*: (Budnik & Avrithis, 2021); HVS: (Duggal et al., 2021); LCE: (Meng et al., 2021).

**Rank weight $W$ in Eq. (9)**. Tab. 2 shows the impact of different ranking weight $W$. When no weight is used, *i.e.*, $W_i = 1$, it leads to unsatisfactory results, and the best results are obtained when both similarity score $S_i$ and ranking position $i$ are considered, simultaneously. Since the images at bottom of the ranking list are more likely dissimilar from the queries, keeping the order between them wastes the representation capability of the query model. It should focus more on the order of samples at the top of the ranking list.

**Various query and gallery models**. In Fig. 5, we study the adaptation to different model architectures. Specifically, models with different architectures are adopted as gallery and query models. See App. A.1 for the number of parameter and computation complexity of different models in details. Our methods outperform CSD in all settings, demonstrating the superiority of rank preserving to the strict neighbor structure alignment.

**Impact of the mapping function** $f(x)$. In Tab. 3, we study three different types of mapping functions introduced in Sec. 4.2. The specific parameters of each function are present in App. C.2. We further visualize the distribution of the similarity scores obtained from three mapping functions in App. D.3. The logarithmic function fits the distribution best and thus leads to the optimal asymmetric retrieval accuracy.

| $f(x)$ | SfM-120k val | INSTRE | $\mathcal{R}$Oxf + $\mathcal{R}$1M Med | $\mathcal{R}$Oxf + $\mathcal{R}$1M Hard | $\mathcal{R}$Par + $\mathcal{R}$1M Med | $\mathcal{R}$Par + $\mathcal{R}$1M Hard |
|---|---|---|---|---|---|---|
| $a^{x-1}$ | 77.83 | **62.69** | 43.32 | 20.51 | 47.94 | 20.66 |
| $\log_a(x+1)$ | **78.94** | 60.61 | **44.24** | **20.93** | **48.13** | **21.45** |
| $\sum_{i=1}^{6} a_i x^{i/2}$ | 78.49 | 60.86 | 43.47 | 20.20 | 47.56 | 20.33 |
| $\sum_{i=1}^{9} a_i x^{i/3}$ | 78.07 | 59.17 | 42.20 | 19.45 | 47.84 | 20.34 |

Table 3: Analysis of **different mapping functions**. SfM-120k, MobileNetV2 and GeM are adopted as training set, query and gallery models, respectively.

**Training with unlabeled data**. Since our methods require no annotations from training set, various unlabeled data can be utilized. In practical scenarios, the deployed gallery images can be utilized to train lightweight query models. Notably, query images, which are unknown in advance, cannot

| Method | Query Net $\phi_q(\cdot)$ | Gallery Net $\phi_g(\cdot)$ | $\mathcal{R}$Oxf Medium | $\mathcal{R}$Oxf Hard | $\mathcal{R}$Oxf + $\mathcal{R}$1M Medium | $\mathcal{R}$Oxf + $\mathcal{R}$1M Hard | $\mathcal{R}$Par Medium | $\mathcal{R}$Par Hard | $\mathcal{R}$Par + $\mathcal{R}$1M Medium | $\mathcal{R}$Par + $\mathcal{R}$1M Hard |
|---|---|---|---|---|---|---|---|---|---|---|
| *Performances of the gallery models* | | | | | | | | | | |
| GeM$^{\ddagger}$ (Radenović et al., 2018b) | Mv2 | Mv2 | 58.81 | 33.41 | 40.02 | 17.71 | 67.87 | 40.97 | 42.25 | 16.59 |
| GeM$^{\ddagger}$ (Radenović et al., 2018b) | R101 | R101 | **65.43** | **40.13** | **45.23** | **19.92** | **76.75** | **55.24** | **52.34** | **24.77** |
| DELG$^{\ddagger}$ (Cao et al., 2020) | Mv2 | Mv2 | 62.42 | 36.56 | 42.21 | 18.64 | 77.91 | 57.96 | 55.09 | 28.81 |
| DELG$^{\ddagger}$ (Cao et al., 2020) | R101 | R101 | **78.55** | **60.89** | **66.02** | **41.75** | **88.58** | **76.05** | **73.65** | **51.46** |
| *Training with **GeM** as gallery model and **SfM-120k** as training set* | | | | | | | | | | |
| Contr* (Budnik & Avrithis, 2021) | | | 47.10 | 21.80 | 18.00 | 6.20 | 61.50 | 37.70 | 28.80 | 8.80 |
| Reg (Budnik & Avrithis, 2021) | | | 49.20 | 23.20 | 26.50 | 7.80 | 65.00 | 40.70 | 34.60 | 12.70 |
| CSD (Wu et al., 2022b) | Mv2 | R101 | 64.12 | 37.53 | 39.38 | 17.73 | 76.16 | 54.29 | 44.40 | 18.08 |
| **Ours** | | | 64.33 | **39.65** | 42.33 | 19.72 | **76.33** | 54.65 | **47.78** | **20.69** |
| **Ours*** | | | **65.22** | 39.50 | **43.47** | **20.20** | 76.31 | **54.82** | 47.56 | 20.33 |
| *Training with **DELG** as gallery model and **GLDv2** as training set* | | | | | | | | | | |
| Reg (Budnik & Avrithis, 2021) | | | 72.75 | 53.07 | 56.03 | 32.21 | 85.81 | 69.96 | 65.23 | 39.29 |
| HVS (Duggal et al., 2021) | | | 74.39 | 54.68 | 58.24 | 34.77 | 86.86 | 72.42 | 67.44 | 43.39 |
| LCE (Meng et al., 2021) | Mv2 | R101 | 75.45 | 54.95 | 58.03 | 33.88 | 87.24 | 73.03 | 67.30 | 43.01 |
| CSD (Wu et al., 2022b) | | | 76.01 | 57.61 | 58.42 | 36.59 | 87.55 | 74.82 | 69.24 | 45.68 |
| **Ours** | | | **79.11** | **59.44** | **64.34** | 39.19 | **89.08** | **76.78** | **72.00** | 48.47 |
| **Ours*** | | | 77.55 | 58.54 | 64.18 | **40.19** | 88.74 | 76.04 | 71.04 | **48.65** |

Table 6: **comparison (asymmetric retrieval) to the state-of-the-art methods.** $^{\ddagger}$: the same gallery models as comparison methods; R101: ResNet101; Mv2: MobileNetV2; **Black bold**: the best performance. See App. B for more comparisons.

participate in the training. In Tab. 4, compared with two other unsupervised algorithms Reg and CSD, our methods achieve the optimal performance. Besides, it shows that when evaluating on a test set, the model trained with the corresponding gallery images leads to better performance. Except for the experiments present in Tab. 4, there is no experiment in the paper, which uses the gallery images from the testing sets to participate in the training.

## 5.3 COMPARISON TO THE STATE-OF-THE-ART METHODS

We compare our method to the state-of-the-art methods in Tab. 5 and Tab. 6. First, we observe that methods based on feature imitation, *e.g.*, Reg, give inferior asymmetric retrieval performance. CSD achieves better results, owing to taking feature preservation and neighbor structure alignment into consideration, simultaneously. Our methods maintain the order of the returned images in ranking list, which is directly related to user experience. It achieves the best performance on all evaluation datasets under the asymmetric setting. When trained with GeM as gallery model and SfM-120k as training set, our framework outperforms the best previous method CSD in mAP by 4.09%, 2.47% on $\mathcal{R}$Oxf + $\mathcal{R}$1M and 3.16%, 2.25% on $\mathcal{R}$Par + $\mathcal{R}$1M, with Medium and Hard protocols, respectively. The results on INSTRE and GLDv2-Test also confirm the superiority of our methods.

## 6 CONCLUSIONS

In this paper, we present a general rank preserving framework for asymmetric image retrieval. Different from strictly feature imitation and neighbor structure alignment, we focus on preserving the order of the returned images in the ranking list when the same query is utilized for symmetric and asymmetric retrieval, respectively. To this end, we devise two instantiations. One is directly constraining the consistency of the sorting results. To make sorting differentiable, sigmoid function is introduced as the smooth approximation for the non-differentiable Heaviside step function used in sorting. The other aims to preserve a monotonic relationship between the returned similarity scores of symmetric and asymmetric retrieval. It introduces a learnable monotonically increasing function to the similarity scores of the symmetric retrieval, which is considered the target of the asymmetric similarity scores. The proposed framework requires no annotation or labels during training, which shows broad applicability and great generalizability in our extensive experiments.

## 7 ACKNOWLEDGMENTS

This work was supported in part by the National Natural Science Foundation of China under Contract 62102128 and 62021001, and in part by the Fundamental Research Funds for the Central Universities under contract WK3490000007. It was also supported by the GPU cluster built by MCC Lab of Information Science and Technology Institution, USTC.

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

# Appendix

In this appendix, we firstly present the pseudo-code of our methods, more details about training, testing and model architectures (App. A). Then, we conduct more comparisons (App. B) to demonstrate the superiority of our methods in various settings. Finally, more extended ablations (App. C) are conducted for an in-depth understanding of our framework (App. D).

---

**Algorithm 1** Pseudo-code of Rank Preserving Framework in a PyTorch-like style.

---

```
# gallery_model: well-trained and fixed encoder for gallery set, no gradient
# query_model: lightweight query encoder, with gradient
# topk: the length of ranking list, K in Equ.(5)
# gallery_feats: features of the training gallery set, embedded by gallery model

for x in training_loader: # load a mini-batch x with N samples from training set
    with torch.no_grad(): # Symmetric retrieval, Sec. 4
        gallery_feat = gallery_model.forward(x) # tensor shape: NxD
        gallery_sim = einsum('bc,kc->bk', gallery_feat, gallery_feats)
        gallery_sim, topk_index = topk(gallery_sim, topk, dim=-1)

    # Asymmetric retrieval, Sec. 4
    query_feat = query_model.forward(x)
    query_sim = einsum('bc,kc->bk', query_feat, gallery_feats)
    query_sim = gather(query_sim, dim=-1, topk_index) # \bm{S}_g shape: Nxtopk

    # two instantiations of the rank preserving framework
    loss = rank_order_preservation(gallery_sim, query_sim)
    or loss = nonlinear_similarity_preservation(gallery_sim, query_sim)

    # SGD update: query model
    loss.backward()
    update(query_model.params)

def sigmoid(x, tau):
    y = 1.0 / (1.0 + exp(-x / tau))
    return y

def rank_order_preservation(sim_g, sim_q): # Rank Order Preservation
    # Indicator matrix \bm{I}_g, Equ. (8)
    sim_g_repeat = sim_g.unsqueeze(1).repeat(1, topk, 1)
    sim_g_diff = sim_g_repeat - sim_g_repeat.permute(0, 2, 1)

    # pass through the sigmoid
    I_g = sigmoid(sim_g_diff, 0.000001) # tesnor shape: Bxtopkxtopk

    # rank weights \bm{W}
    sim_weight = softmax(sim_g * tau_r, dim=-1) # tensor shape : Bxtopk
    position_weight = arange(start=0, end=topk) + 1.0 # tensor shape : Bxtopk
    rank_weight = (sim_weight * position_weight).unsqueeze(-1)

    # Indicator matrix \bm{I}_q, Equ (8)
    sim_q_repeat = sim_q.unsqueeze(1).repeat(1, topk, 1)
    sim_q_diff = sim_q_repeat - sim_g_repeat.permute(0, 2, 1)

    # pass through the sigmoid
    I_q = sigmoid(sim_q_diff, tua) # tua: \tua in Equ. (10)
    return sum(pow(I_g - I_q,2) * rank_weight,dim=(-1,-2)).mean()

def monotonic_similarity_preservation(sim_g, sim_q): # Monotonic Similarity
    Preservation
    sim_g_mapped = f(sim_g) # f(x) is the mapping function in Sec. 4.2
    p_g = softmax(sim_g_mapped * tau_g, dim=-1) # tensor shape: Bxtopk
    p_q = softmax(sim_q * tau_g, dim=-1).log() # tensor shape: Bxtopk
    return kl_div(p_q, p_g, reduction='mean')
```

---

einsum: einstein summation convention; gather: gather values along an axis; cat: concatenation; softmax: softmax function.

---

| Query | Gallery | FLOPs (G) | | Param. (M) | |
| Net $\phi_q(\cdot)$ | Net $\phi_g(\cdot)$ | ABS | % | ABS | % |
|---|---|---|---|---|---|
| ResNet101 | ResNet101 | 42.85 | 100.0 | 42.50 | 100.0 |
| ShuffleNetV2 (0.5×) | | 0.84 | 1.96 | 2.44 | 5.74 |
| ShuffleNetV2 | | 1.44 | 3.36 | 3.35 | 7.88 |
| MobileNetV2 | | 2.50 | 5.83 | 4.85 | 11.41 |
| EfficientNetB0 | ResNet101 | 2.86 | 6.67 | 6.63 | 15.60 |
| EfficientNetB1 | | 3.92 | 9.15 | 9.13 | 21.49 |
| EfficientNetB2 | | 4.50 | 10.51 | 10.58 | 24.90 |
| EfficientNetB3 | | 6.24 | 14.57 | 13.84 | 32.56 |

Table 7: **The number of parameters and the computational complexity (in FLOPs)** of query models used in this work. To adapt for image retrieval, all layers except feature extractor are removed. A convolutional layer is added to the feature extractor for matching the output dimension of the gallery model. (0.5×) denotes a model with 0.5× width.



(a) Classification        (b) Image retrieval

Figure 6: **Model architecture** comparison of classification and image retrieval task.

## A  IMPLEMENTATION DETAILS

In Alg. 1, we demonstrate the pseudo-code of our methods. All the training and evaluation datasets are publicly available, and the splits have been carefully described in Sec. 5. Training and testing details are present at App. A.1.

### A.1  EXPERIMENT SETTINGS

**Training Details**. When SfM-120k is adopted for training, we follow the common settings in AML (Budnik & Avrithis, 2021). Training images are resized with the maximum side equal to 512, keeping the aspect ratio. No data augmentation is adopted. Training epochs and batch size are set as 10 and 64, respectively. The query model is trained on one NVIDIA RTX 3090 GPU. When using GLDv2 as the training set, $512 \times 512$ pixels are center cropped from the randomly resized image. Random color jittering is adopted as the data augmentation. We train the query model on 4 NVIDIA RTX 3090 GPUs for 10 epochs with a batch size of 256. All models are optimized using Adam with an initial learning rate of $10^{-3}$ and a weight decay of $10^{-6}$. A linearly decaying scheduler is adopted to gradually decay the learning rate to 0 when the desired number of steps is reached. When query model is trained with **Rank Order Preservation** (Sec. 4.1), the length $K$ of ranking list is set to 4096, and the temperature coefficient $\tau_r$ in ranking weight $W_i$ is set as 0.2. As for **Monotonic Similarity Preservation** (Sec. 4.2), $K$ is also set as 4096, and both $\tau_g$ and $\tau_q$ are set to 0.1. The gallery set $\mathcal{G}_t$, used during training, is the same as training dataset in all cases.

**Details about Testing Datasets**. GLDv2-Test contains $761,757$ gallery images and 750 queries, with evaluation metric of mAP@100. As for $\mathcal{R}$Oxf and $\mathcal{R}$Par, there are 70 queries for both of them, with $4,993$ and $5,007$ gallery images, respectively. mAP on the Medium (Med) and Hard settings are reported. Large-scale results are reported with the $\mathcal{R}$1M (Radenović et al., 2018a) dataset added, which contains 1M distractor images. The last corpus is INSTRE (Wang & Jiang, 2015), which contains various everyday 3D or planar objects from buildings to book covers in natural scenes. We follow the data partitioning proposed by Iscen et al. (2017) with $1,250$ queries and $27,293$ gallery images. mAP is reported with another 1M distractor images $\mathcal{D}$1M added into the gallery or not.

**Model Architecture Details**. In Fig. 6, we show the typical differences in the model architecture for the classification task and image retrieval task. To obtain better transfer performance, most of the models are pre-trained on ImageNet (Deng et al., 2009) with category labels. These models usually consist of a feature extractor, a global mean pooling, several fully-connected layers, and a classification layer. To adapt these models for the image retrieval task, we keep only the feature extractor and discard the other layers. After that, GeM pooling (Radenović et al., 2018b) is adopted for aggregating the feature map output by the feature extractor. Finally, a whitening layer, implemented by a fully-connected layer, is adopted to obtain the final global feature. Notably, the whitening layer

| Method | Query NET $\phi_q(\cdot)$ | Gallery NET $\phi_g(\cdot)$ | $\mathcal{R}$Oxf Medium | $\mathcal{R}$Oxf Hard | $\mathcal{R}$Oxf + $\mathcal{R}$1M Medium | $\mathcal{R}$Oxf + $\mathcal{R}$1M Hard | $\mathcal{R}$Par Medium | $\mathcal{R}$Par Hard | $\mathcal{R}$Par + $\mathcal{R}$1M Medium | $\mathcal{R}$Par + $\mathcal{R}$1M Hard |
|---|---|---|---|---|---|---|---|---|---|---|
| *Performances of the gallery models* | | | | | | | | | | |
| GeM$^{\ddagger}$ (Radenović et al., 2018b) | Eb3 | Eb3 | 54.22 | 27.53 | 37.10 | 17.49 | 71.21 | 48.00 | 44.67 | 18.45 |
| GeM$^{\ddagger}$ (Radenović et al., 2018b) | R101 | R101 | **65.43** | **40.13** | **45.23** | **19.92** | **76.75** | **55.24** | **52.34** | **24.77** |
| DELG$^{\ddagger}$ (Cao et al., 2020) | Eb3 | Eb3 | 66.64 | 43.82 | 49.67 | 24.89 | 81.78 | 63.90 | 61.10 | 32.34 |
| DELG$^{\ddagger}$ (Cao et al., 2020) | R101 | R101 | **78.55** | **60.89** | **66.02** | **41.75** | **88.58** | **76.05** | **73.65** | **51.46** |
| *Training with **GeM** as gallery model and **SfM-120k** as training set* | | | | | | | | | | |
| Contr* (Budnik & Avrithis, 2021) | | | 45.20 | 19.60 | 24.70 | 12.20 | 63.70 | 40.90 | 32.80 | 12.50 |
| Reg (Budnik & Avrithis, 2021) | | | 52.90 | 27.80 | 29.70 | 10.40 | 65.20 | 42.40 | 39.00 | 16.00 |
| CSD (Wu et al., 2022b) | | | 65.16 | 38.62 | 43.05 | 18.81 | 75.94 | 53.05 | 46.76 | 19.43 |
| **Ours** | Eb3 | R101 | 65.44 | 39.36 | 45.33 | **21.50** | **76.35** | 54.77 | 52.07 | 24.10 |
| *(mAP gains over CSD)* | | | (↑ 0.28) | (↑ 0.74) | (↑ 2.28) | (↑ 2.69) | (↑ 0.41) | (↑ 1.72) | (↑ 5.31) | (↑ 4.67) |
| **Ours*** | | | **66.26** | **39.83** | **46.20** | 21.39 | 76.33 | 53.68 | 50.65 | 22.97 |
| *(mAP gains over CSD)* | | | (↑ 1.10) | (↑ 1.21) | (↑ 3.15) | (↑ 2.58) | (↑ 0.39) | (↑ 0.53) | (↑ 3.89) | (↑ 3.54) |
| *Training with **DELG** as gallery model and **GLDv2** as training set* | | | | | | | | | | |
| Reg (Budnik & Avrithis, 2021) | | | 74.60 | 53.41 | 59.88 | 33.31 | 83.81 | 68.15 | 59.36 | 35.24 |
| HVS (Duggal et al., 2021) | | | 76.41 | 56.13 | 62.72 | 36.86 | 87.07 | 74.53 | 71.54 | 49.09 |
| LCE (Meng et al., 2021) | | | 75.89 | 55.21 | 61.90 | 36.53 | 86.63 | 73.62 | 70.98 | 48.94 |
| CSD (Wu et al., 2022b) | Eb3 | R101 | 77.44 | 58.97 | 63.21 | 38.20 | 87.94 | 75.68 | 73.37 | 50.09 |
| **Ours** | | | 80.23 | **62.60** | 65.40 | **40.93** | 89.35 | **77.35** | 73.45 | 50.33 |
| *(mAP gains over CSD)* | | | (↑ 2.79) | (↑ 3.63) | (↑ 2.19) | (↑ 2.73) | (↑ 1.41) | (↑ 1.67) | (↑ 0.08) | (↑ 0.24) |
| **Ours*** | | | **80.60** | 62.45 | **66.21** | 40.86 | **89.36** | 76.78 | **74.24** | **51.05** |
| *(mAP gains over CSD)* | | | (↑ 3.16) | (↑ 3.48) | (↑ 3.00) | (↑ 2.66) | (↑ 1.42) | (↑ 1.10) | (↑ 0.87) | (↑ 0.94) |

Table 8: **mAP (asymmetric retrieval) comparison to the state-of-the-art methods.** DELG and GeM are trained with SfM-120k and GLDv2, respectively. $^{\ddagger}$: gallery models are the same as comparison methods; R101: ResNet101; Eb3: EfficientNetB3. **Black bold** denotes the best performance.

is pre-trained in the embedding space of the gallery model and remains fixed during the training of the query model. In Tab. 7, we show the number of parameters and computational complexity (in FLOPS) of different networks, all modified for the retrieval task, when the size of the input image is $362 \times 362$. Compared with large models, lightweight models significantly reduce the computation during inference phase. Thus, they can be used in various resource-constrained scenarios.

**Testing Details**. During testing, as for $\mathcal{R}$Oxf and $\mathcal{R}$Par datasets, we resize images so that the larger dimension is equal to 1024 pixels and preserve the aspect ratio. Besides, the image features are extracted at three scales, *i.e.*, $\{1/\sqrt{2}, 1, \sqrt{2}\}$. We perform $L_2$ normalization for each scale independently and the features of three scales are averaged, followed by another $L_2$ normalization. Under the *asymmetric retrieval* setting, queries are embedded with the lightweight query model $\phi_q(\cdot)$, while the gallery images are embedded by a large model $\phi_g(\cdot)$.

## B   EXTENDED COMPARISONS

### B.1   DIFFERENT QUERY MODEL

In this section, we perform more extensive comparisons with existing methods. Specifically, in Tab. 8, we take EfficientNetB3 as the lightweight query model. The performance of asymmetric retrieval improves as the capacity of the query model becomes larger (compared with the accuracy of MobileNetV2 in Tab. 6). Our proposed methods achieve optimal performance in various settings, with the performance of asymmetric retrieval being almost comparable to that of symmetric retrieval.

### B.2   DIFFERENT GALLERY MODEL

In Tab. 9, we further extend the experiments in Fig. 5 in the main paper. Three recent deep representation models are adopted to embed gallery images. (1) Token (Wu et al., 2022a) based on local feature aggregation. (2) CVNet (Lee et al., 2022) based on global pooling. (3) DOLG (Yang et al., 2021) based on local and global feature fusion. Our methods achieve the best performance across various settings, which demonstrates the generalization of them.

| Method | Query NET $\phi_q(\cdot)$ | Gallery NET $\phi_g(\cdot)$ | $\mathcal{R}$Oxf | | $\mathcal{R}$Oxf + $\mathcal{R}$1M | | $\mathcal{R}$Par | | $\mathcal{R}$Par + $\mathcal{R}$1M | |
|---|---|---|---|---|---|---|---|---|---|---|
| | | | Medium | Hard | Medium | Hard | Medium | Hard | Medium | Hard |
| *Performances of the gallery models* | | | | | | | | | | |
| Token[‡] (Wu et al., 2022a) | | | 82.16 | 65.75 | 70.58 | 47.46 | 89.40 | 78.44 | 77.24 | 56.81 |
| CVNet[‡] (Lee et al., 2022) | R101 | R101 | 80.01 | 62.83 | 74.25 | 54.56 | 90.18 | 79.01 | 80.82 | 62.74 |
| DOLG[‡] (Yang et al., 2021) | | | 82.37 | 64.94 | 75.19 | 53.55 | 90.97 | 81.71 | 82.28 | 66.45 |
| *Traing with **Token** as gallery model* | | | | | | | | | | |
| Reg[†] (Budnik & Avrithis, 2021) | | | 71.67 | 50.92 | 59.55 | 35.41 | 83.26 | 69.48 | 63.98 | 42.93 |
| HVS[†] (Duggal et al., 2021) | | | 73.16 | 53.19 | 57.68 | 35.52 | 84.27 | 71.43 | 62.90 | 41.00 |
| LCE[†] (Meng et al., 2021) | Mv2 | R101 | 74.82 | 56.40 | 61.57 | 39.43 | 84.09 | 71.88 | 65.70 | 43.86 |
| CSD[†] (Wu et al., 2022b) | | | 75.52 | 56.83 | 63.46 | 39.01 | 84.50 | 70.73 | 65.93 | 43.77 |
| **Ours** | | | **77.22** | 57.33 | 65.87 | 40.97 | 85.51 | 71.95 | 67.88 | 44.87 |
| **Ours*** | | | 76.99 | **57.35** | **66.00** | **41.24** | **86.35** | **73.69** | **70.37** | **48.89** |
| *Training with **CVNet** as gallery model* | | | | | | | | | | |
| Reg[†] (Budnik & Avrithis, 2021) | | | 74.60 | 53.41 | 59.88 | 33.31 | 83.81 | 68.15 | 59.36 | 35.24 |
| HVS[†] (Duggal et al., 2021) | | | 74.90 | 55.17 | 62.32 | 42.66 | 84.92 | 71.62 | 67.13 | 46.80 |
| LCE[†] (Meng et al., 2021) | Mv2 | R101 | 75.95 | 57.87 | 63.77 | 43.37 | 83.66 | 69.71 | 66.44 | 46.72 |
| CSD[†] (Wu et al., 2022b) | | | 76.44 | 58.41 | 64.42 | 43.90 | 85.32 | 71.61 | 68.32 | 47.76 |
| **Ours** | | | 76.62 | 59.48 | 66.49 | 46.26 | 86.48 | 73.58 | 72.89 | 53.11 |
| **Ours*** | | | **77.92** | **60.83** | **68.68** | **47.04** | **87.81** | **75.62** | **75.16** | **55.97** |
| *Training with **DOLG** as gallery model* | | | | | | | | | | |
| Reg[†] (Budnik & Avrithis, 2021) | | | 69.04 | 48.00 | 56.81 | 35.51 | 79.13 | 63.13 | 60.22 | 42.59 |
| HVS[†] (Duggal et al., 2021) | | | 72.79 | 54.20 | 63.29 | 41.74 | 85.18 | 70.72 | 68.13 | 48.25 |
| LCE[†] (Meng et al., 2021) | Mv2 | R101 | 72.84 | 53.70 | 61.90 | 40.84 | 85.77 | 69.54 | 67.65 | 48.53 |
| CSD[†] (Wu et al., 2022b) | | | 75.53 | 56.23 | 64.02 | 42.79 | 86.34 | 72.84 | 69.29 | 49.47 |
| **Ours** | | | 77.67 | 59.87 | **67.40** | **44.93** | 87.19 | 73.64 | 70.91 | 52.38 |
| **Ours*** | | | **78.46** | **60.74** | 66.98 | 43.84 | **87.81** | **74.72** | **72.18** | **53.63** |

Table 9: **Extend mAP (asymmetric retrieval) comparison** to the state-of-the-art methods with **different gallery models**. All the models are trained with GLDv2. [‡]: re-evaluate the official public weights; [†]: our re-implementation. R101: ResNet101; Mv2: MobileNetV2. **Black bold** denotes the best performance.

## C   ADDITIONAL ABLATIONS

### C.1   IMPACT OF DISTANCE TYPE

In **Rank Order Preservation** (Sec. 4.1), the weighted mean square error is token as the final objective function to train the query model end to end. In the section, we explore an alternative to measure the distance between the Indicator matrices $\boldsymbol{I}^g$ and $\boldsymbol{I}^q$. Specifically, we convert each row of $\boldsymbol{I}^g$ and $\boldsymbol{I}^q$ into the form of probability distribution:

$$\mathbf{P}^g_{i,j} = \frac{\exp\left(\boldsymbol{I}^g_{i,j}/\tau_g\right)}{\sum_{l=1}^K \exp\left(\boldsymbol{I}^g_{i,l}/\tau_g\right)}, \mathbf{P}^q_{i,j} = \frac{\exp\left(\boldsymbol{I}^q_{i,j}/\tau_q\right)}{\sum_{l=1}^K \exp\left(\boldsymbol{I}^q_{i,l}/\tau_q\right)}, i = 1, 2, \cdots, K, \tag{12}$$

where $\tau_g$ and $\tau_q$ are the temperatures. Then, the distance between two indicator matrices is defined as the KL divergence of two distributions:

$$\mathcal{L}_{\mathrm{ROP}} = \sum_{i=1}^K \sum_{j=1}^K \boldsymbol{W}_i \mathbf{P}^g_{i,j} \log\left(\frac{\mathbf{P}^g_{i,j}}{\mathbf{P}^q_{i,j}}\right). \tag{13}$$

As for **Monotonic Similarity Preservation** (Sec. 4.2), we try to use the $L_2$ distance to measure the inconsistency between the mapped similarity scores $\boldsymbol{M}_g$ and the asymmetric similarity scores $\boldsymbol{S}_q$. Then, Eq. (11) is defined as

$$\mathcal{L}_{\mathrm{MSP}} = \|\boldsymbol{M}_g - \boldsymbol{S}_q\|_2^2 = \sum_{i=1}^K (\boldsymbol{M}^i_g - \boldsymbol{S}^i_q)^2. \tag{14}$$

| Gallery Net $\phi_g(\cdot)$ | Loss function | GLDv2-Test | INSTRE $+\varnothing$ | INSTRE $+\mathcal{D}1M$ | $\mathcal{R}$Oxf + $\mathcal{R}$1M Med | $\mathcal{R}$Oxf + $\mathcal{R}$1M Hard | $\mathcal{R}$Par + $\mathcal{R}$1M Med | $\mathcal{R}$Par + $\mathcal{R}$1M Hard |
|---|---|---|---|---|---|---|---|---|
| GeM | Eq. (12) | **13.33** | 59.83 | **50.98** | **42.49** | **19.99** | **48.05** | 20.15 |
| | Eq. (9) | 12.35 | **60.81** | 50.96 | 42.33 | 19.78 | 47.72 | **20.34** |
| DELG | Eq. (12) | **24.92** | **35.67** | **25.05** | 64.23 | **39.65** | 70.90 | 48.18 |
| | Eq. (9) | 23.86 | 29.93 | 21.76 | **64.34** | 39.19 | **72.00** | **48.47** |

(a) Query model $\phi_q(\cdot)$ is trained by **Rank Order Preservation** (Sec. 4.1).

| Gallery Net $\phi_g(\cdot)$ | Loss function | GLDv2-Test | INSTRE $+\varnothing$ | INSTRE $+\mathcal{D}1M$ | $\mathcal{R}$Oxf + $\mathcal{R}$1M Med | $\mathcal{R}$Oxf + $\mathcal{R}$1M Hard | $\mathcal{R}$Par + $\mathcal{R}$1M Med | $\mathcal{R}$Par + $\mathcal{R}$1M Hard |
|---|---|---|---|---|---|---|---|---|
| GeM | Eq. (14) | 12.79 | **60.95** | **52.48** | 43.00 | **20.83** | **48.46** | **20.63** |
| | Eq. (11) | **13.33** | 60.86 | 49.72 | **43.47** | 20.20 | 47.56 | 20.33 |
| DELG | Eq. (14) | 25.02 | **35.21** | **27.50** | **64.75** | **42.18** | **71.13** | **48.74** |
| | Eq. (11) | **25.44** | 34.34 | 26.58 | 64.18 | 40.19 | 71.04 | 48.65 |

(b) Query model $\phi_q(\cdot)$ is trained by **Monotonic Similarity Preservation** (Sec. 4.2).

Table 10: mAP (asymmetric retrieval) comparison of different **distance types**. SfM-120k and GLDv2 are adopted for training the query model when GeM and DELG serve as the gallery model, respectively. MobileNetV2 is adopted as the query model.

| Map func. | SfM-120k val | INSTRE | $\mathcal{R}$Oxf + $\mathcal{R}$1M Med | $\mathcal{R}$Oxf + $\mathcal{R}$1M Hard | $\mathcal{R}$Par + $\mathcal{R}$1M Med | $\mathcal{R}$Par + $\mathcal{R}$1M Hard |
|---|---|---|---|---|---|---|
| *Logarithmic function* | | | | | | |
| $\log_{2.2}(x+1)$ | 78.26 | 57.90 | 41.31 | 18.52 | 46.13 | 19.21 |
| $\log_{2.4}(x+1)$ | 78.12 | 60.19 | 42.84 | 19.59 | 47.83 | 19.89 |
| $\log_e(x+1)$ | **78.94** | **60.61** | **44.24** | 20.93 | 48.13 | **21.45** |
| $\log_3(x+1)$ | 77.98 | 60.05 | 42.60 | 19.61 | 48.49 | 20.38 |
| $\dagger\log_{2.99}(x+1)$ | 78.24 | 60.49 | 43.01 | 20.51 | **48.52** | 20.69 |
| *Exponential function* | | | | | | |
| $6^{x-1}$ | 76.59 | 61.17 | 41.79 | 19.32 | 46.29 | 19.39 |
| $8^{x-1}$ | 77.55 | 62.21 | 43.08 | 19.77 | 47.78 | 19.89 |
| $10^{x-1}$ | **77.83** | **62.69** | **43.32** | 20.51 | 47.94 | **20.66** |
| $12^{x-1}$ | 77.61 | 62.63 | 43.20 | 20.14 | 47.82 | 20.48 |
| $\dagger 10.69^{x-1}$ | 77.47 | 62.15 | 43.13 | 19.72 | 47.76 | 20.20 |

Table 11: Ablation on **different mapping functions**. † denotes that the base is learned from the data distribution, otherwise is determined artificially in advance. MobileNetV2 and GeM are adopted as query and gallery models, respectively.

| Map func. | SfM-120k val | INSTRE | $\mathcal{R}$Oxf + $\mathcal{R}$1M Med | $\mathcal{R}$Oxf + $\mathcal{R}$1M Hard | $\mathcal{R}$Par + $\mathcal{R}$1M Med | $\mathcal{R}$Par + $\mathcal{R}$1M Hard |
|---|---|---|---|---|---|---|
| *Polynomial function* | | | | | | |
| $\sum_{i=1}^{2} a_i x^{i/2}$ | 75.65 | 54.55 | 39.88 | 18.08 | 44.87 | 18.34 |
| $\sum_{i=1}^{4} a_i x^{i/2}$ | 78.05 | 58.21 | 42.13 | 19.76 | 46.74 | 19.56 |
| $\sum_{i=1}^{6} a_i x^{i/2}$ | **78.49** | **60.86** | **43.47** | **20.20** | **47.56** | **20.33** |
| $\sum_{i=1}^{3} a_i x^{i/3}$ | 76.19 | 56.39 | 38.27 | 17.40 | 43.92 | 17.76 |
| $\sum_{i=1}^{6} a_i x^{i/3}$ | 77.47 | 58.05 | 42.41 | 19.31 | 46.16 | 19.85 |
| $\sum_{i=1}^{9} a_i x^{i/3}$ | **78.07** | **59.17** | **42.20** | **19.45** | **47.84** | **20.34** |
| $\sum_{i=1}^{4} a_i x^{i/4}$ | 76.63 | 53.99 | 40.25 | 17.76 | 44.23 | 17.62 |
| $\sum_{i=1}^{8} a_i x^{i/4}$ | 77.22 | 58.33 | 41.02 | 18.29 | 45.93 | 18.81 |
| $\sum_{i=1}^{12} a_i x^{i/4}$ | **77.94** | **59.02** | **42.18** | **19.84** | **46.43** | **19.32** |

Table 12: Ablation on **different mapping functions**. Polynomial function is adopted as the mapping function. MobileNetV2 and GeM are adopted as query and gallery models, respectively.

The comparison of different distance type are summarized in Tab. 10, the performance of various loss types are similar, which shows that rank preserving is the key to achieve superior performance rather than a specific consistency loss.

## C.2 IMPACT OF THE UPDATE FUNCTION

In Tab. 11 and Tab. 12, we show an extend version of Tab. 3 in the main paper, where the specific parameter values are given. As for the logarithmic and exponential functions, we also consider manually defining the bases of the mapping functions. As shown in Tab. 11, both learnable bases and manual defined bases lead to similar results. For the polynomial functions, we consider various settings of the basis functions, *i.e.*, $\{x^{1/2\alpha}\}_{\alpha=1}^{6}, \{x^{1/3\alpha}\}_{\alpha=1}^{9}, \{x^{1/4\alpha}\}_{\alpha=1}^{12}$. Note that the parameters $\{a_1, a_2, \cdots, a_N\}$ in this case are all learned from the data distribution. It can be seen that superior performance is achieved with various settings of the basis functions when the order of the basis functions is properly chosen.

## C.3 EXTENDED ABLATION ON TRAINING DATASETS

In this section, we study the scalability of our framework. we randomly sample different number of images from GLDv2 dataset for training. As shown in Tab. 13, more training data leads to better

| Gallery Net $\phi_g(\cdot)$ | Training Set $\mathcal{T}$ | Image Numbers | GLDv2-Test | INSTRE +$\varnothing$ | +$\mathcal{D}$1M | $\mathcal{R}$Oxf + $\mathcal{R}$1M Med | Hard | $\mathcal{R}$Par + $\mathcal{R}$1M Med | Hard |
|---|---|---|---|---|---|---|---|---|---|
| GeM | SfM-120k | 91,642 | 12.35 | 60.81 | 50.96 | 39.65 | 19.78 | 47.72 | 20.34 |
| | GLDv2 (×0.1) | 128,078 | 14.16 | 59.29 | 48.75 | 44.24 | 20.18 | 49.64 | 21.93 |
| | GLDv2 (×0.2) | 256,156 | 14.62 | 61.06 | 51.57 | 45.95 | 22.34 | 51.33 | 23.57 |
| | GLDv2 (×0.4) | 512,312 | **14.77** | **62.90** | **54.00** | **47.01** | **23.61** | **53.16** | **25.47** |
| DELG | SfM-120k | 91,642 | 18.62 | 23.89 | 14.49 | 52.08 | 30.54 | 59.63 | 33.86 |
| | GLDv2 (×0.1) | 128,078 | 23.39 | 28.64 | 22.19 | 61.99 | 38.87 | 67.83 | 46.25 |
| | GLDv2 (×0.2) | 256,156 | 24.14 | 29.28 | 22.53 | 62.31 | 39.40 | 68.14 | 46.80 |
| | GLDv2 (×0.4) | 512,312 | **24.79** | **32.48** | **25.70** | **63.23** | **40.23** | **69.40** | **47.21** |

(a) Query model $\phi_q(\cdot)$ is trained by **Rank Order Preservation** (Sec. 4.1).

| Gallery Net $\phi_g(\cdot)$ | Training Set $\mathcal{T}$ | Image Numbers | GLDv2-Test | INSTRE +$\varnothing$ | +$\mathcal{D}$1M | $\mathcal{R}$Oxf + $\mathcal{R}$1M Med | Hard | $\mathcal{R}$Par + $\mathcal{R}$1M Med | Hard |
|---|---|---|---|---|---|---|---|---|---|
| GeM | SfM-120k | 91,642 | 13.33 | 60.86 | 49.72 | 43.47 | 20.20 | 47.56 | 20.33 |
| | GLDv2 (×0.1) | 128,078 | 14.13 | 62.41 | 53.42 | 46.88 | 22.53 | 51.41 | 23.48 |
| | GLDv2 (×0.2) | 256,156 | 14.40 | 63.82 | 55.32 | 47.15 | 23.16 | 52.96 | 25.35 |
| | GLDv2 (×0.4) | 512,312 | **14.86** | **65.26** | **57.48** | **48.90** | **23.97** | **53.17** | **25.91** |
| DELG | SfM-120k | 91,642 | 19.07 | 25.12 | 13.35 | 52.19 | 28.35 | 56.19 | 30.03 |
| | GLDv2 (×0.1) | 128,078 | 24.12 | 32.42 | 24.16 | 60.20 | 37.72 | 67.84 | 45.69 |
| | GLDv2 (×0.2) | 256,156 | 24.27 | 33.02 | 25.34 | 62.31 | 38.66 | 68.23 | 46.41 |
| | GLDv2 (×0.4) | 512,312 | **24.63** | **33.76** | **26.33** | **63.45** | **39.52** | **69.47** | **47.67** |

(b) Query model $\phi_q(\cdot)$ is trained by **Monotonic Similarity Preservation** (Sec. 4.2).

Table 13: mAP (asymmetric retrieval) comparison of different **training dataset size**. We randomly sample some images from the original training datasets to form the new training set ,where (×$x$) denotes the proportion. MobileNetV2 is adopted as the query model $\phi_q(\cdot)$.

| Gallery Net $\phi_g(\cdot)$ | Training Set $\mathcal{T}$ | GLDv2-Test | $\mathcal{R}$Oxf + $\mathcal{R}$1M Med | Hard | $\mathcal{R}$Par + $\mathcal{R}$1M Med | Hard |
|---|---|---|---|---|---|---|
| Token | GLDv2 | **27.12** | 65.87 | 40.97 | 67.88 | 44.87 |
| | $\mathcal{R}$Oxf$^{\dagger}$ + $\mathcal{R}$Par$^{\dagger}$ + $\mathcal{R}$1M | 26.33 | **66.95** | **42.26** | **74.63** | **53.16** |
| CVNet | GLDv2 | **28.68** | 66.49 | 46.26 | 72.89 | 53.11 |
| | $\mathcal{R}$Oxf$^{\dagger}$ + $\mathcal{R}$Par$^{\dagger}$ + $\mathcal{R}$1M | 26.39 | **69.58** | **48.39** | **78.32** | **59.22** |
| DOLG | GLDv2 | **25.16** | 67.40 | 44.93 | 70.91 | 52.38 |
| | $\mathcal{R}$Oxf$^{\dagger}$ + $\mathcal{R}$Par$^{\dagger}$ + $\mathcal{R}$1M | 24.73 | **68.83** | **46.41** | **75.21** | **55.2** |

(a) Query model $\phi_q(\cdot)$ is trained by **Rank Order Preservation** (Sec. 4.1).

| Gallery Net $\phi_g(\cdot)$ | Training Set $\mathcal{T}$ | GLDv2-Test | $\mathcal{R}$Oxf + $\mathcal{R}$1M Med | Hard | $\mathcal{R}$Par + $\mathcal{R}$1M Med | Hard |
|---|---|---|---|---|---|---|
| Token | GLDv2 | **27.55** | 66.00 | 41.24 | 70.37 | 48.89 |
| | $\mathcal{R}$Oxf$^{\dagger}$ + $\mathcal{R}$Par$^{\dagger}$ + $\mathcal{R}$1M | 26.17 | **66.37** | **41.89** | **73.85** | **52.28** |
| CVNet | GLDv2 | **28.57** | **68.68** | 47.04 | 75.16 | 55.97 |
| | $\mathcal{R}$Oxf$^{\dagger}$ + $\mathcal{R}$Par$^{\dagger}$ + $\mathcal{R}$1M | 26.05 | 68.51 | **47.08** | **78.50** | **59.26** |
| DOLG | GLDv2 | **26.25** | 66.98 | 43.84 | 72.18 | 53.63 |
| | $\mathcal{R}$Oxf$^{\dagger}$ + $\mathcal{R}$Par$^{\dagger}$ + $\mathcal{R}$1M | 24.96 | **69.11** | **44.69** | **77.34** | **59.14** |

(b) Query model $\phi_q(\cdot)$ is trained by **Monotonic Similarity Sreservation** (Sec. 4.2).

Table 14: Extended experiments about **training on the deployed gallery set**. The query sets for all test sets are not involved in training. † denotes the gallery set of that testing dataset. MobileNetV2 is adopted as query models.

performance across all settings. In Tab. 13, we conduct more experiments with unlabeled gallery set for training. When evaluating on a deployed database, training on it is always better than on a collected training dataset, which is also confirmed by Tab. 14 in the main paper.

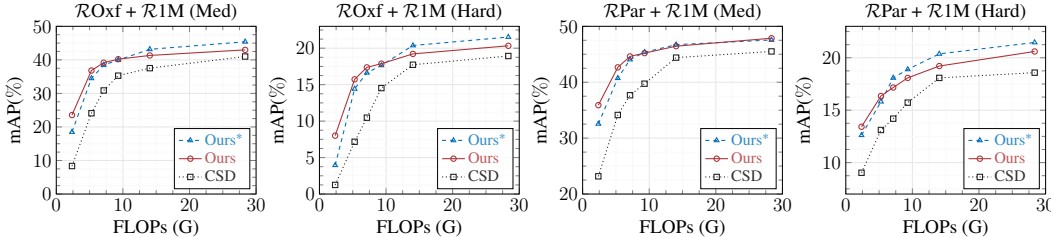

Figure 7: **Performance versus inference computation complexity when varying the size of input image**. MobileNetV2 and GeM are adopted as query and gallery models, respectively. Query features are extracted as the original single scale. The x-axis represents the average FLOPs (G) for five inferences, which is proportional to the size of the image. We resize the queries to $\{0.2, 0.4, 0.6, 1/\sqrt{2}, 0.8, 1.0, \sqrt{2}\}$ of the original size $(1024 \times 768)$.

## D  ANALYSIS AND DISCUSSIONS

### D.1  INFERENCE COMPUTATION

During testing, we follow the common settings to use multi-scale feature extraction, which greatly aggravates the inference computation of the query model. The inference computation is mainly related to the complexity of the model and the size of the test image. In this section, MobileNetV2 is adopted as query model. We use single-scale feature extraction and vary the image size to study the relationship between inference computation and asymmetric retrieval accuracy. The results are shown in Fig. 7, the asymmetric retrieval accuracy increases and saturates as the inference computation increases. In a practical application scenario, we should choose the appropriate test image size to achieve a balance between efficiency and accuracy.

| Gallery Net $\phi_g(\cdot)$ | Loss function | GLDv2-Test | INSTRE $+ \varnothing$ | INSTRE $+ \mathcal{D}1M$ | $\mathcal{R}$Oxf + $\mathcal{R}$1M Med | $\mathcal{R}$Oxf + $\mathcal{R}$1M Hard | $\mathcal{R}$Par + $\mathcal{R}$1M Med | $\mathcal{R}$Par + $\mathcal{R}$1M Hard |
|---|---|---|---|---|---|---|---|---|
| GeM | Eq.(11) | 13.33 | 60.86 | 49.72 | **43.47** | 20.20 | 47.56 | 20.33 |
| | Eq.(9) | 12.35 | 60.81 | 50.96 | 42.33 | 19.78 | 47.72 | 20.34 |
| | $\lambda_1$Eq.(9) + $\lambda_2$Eq.(11) | **12.71** | **61.56** | **53.36** | 43.27 | **20.55** | **48.86** | **20.91** |
| DELG | Eq.(11) | **25.44** | 34.34 | **26.58** | 64.18 | 40.19 | 71.04 | 48.65 |
| | Eq.(9) | 23.86 | 29.93 | 21.76 | 64.34 | 39.19 | **72.00** | **48.47** |
| | $\lambda_1$Eq.(9) + $\lambda_2$Eq.(11) | 24.84 | **34.84** | 26.56 | **64.56** | **40.79** | 71.64 | 48.02 |

Table 15: Ablation on the **combination of different training losses**. SfM-120k and GLDv2 are adopted for training the query model when GeM and DELG serve as the gallery model, respectively. MobileNetV2 is adopted as the query model.

### D.2  COMBINE TWO INSTANTIATIONS

In this section, we try to combine the two instantiation methods proposed in the main paper. As shown in Tab. 15, the simple combination fails to bring further performance improvement. We believe this is due to the fact that the optimization goals of both methods are to maintain the order of the images in the returned ranking list. Thus, the final results obtained with two methods are not complementary to each other, which is also confirmed by the distribution of the similarity scores present in Fig. 3 and Fig. 8.

### D.3  SIMILARITY SCORE DISTRIBUTION

In this section, we visualize the similarity score distributions in Fig. 8, when the query model is trained by different methods. The first row shows the training process of **Rank Order Preservation**. The following three rows correspond to the training process of **Monotonic Similarity Preservation**, when three different mapping functions are chosen, respectively. As the training proceeds, the region of the similarity distributions gradually becomes slender, which indicates that the orders of the images in the returned ranking list are better maintained. It can be found that under our rank

preserving framework, the different instantiations yield similar results, which is also confirmed by the final retrieval accuracy (Tab. 6).

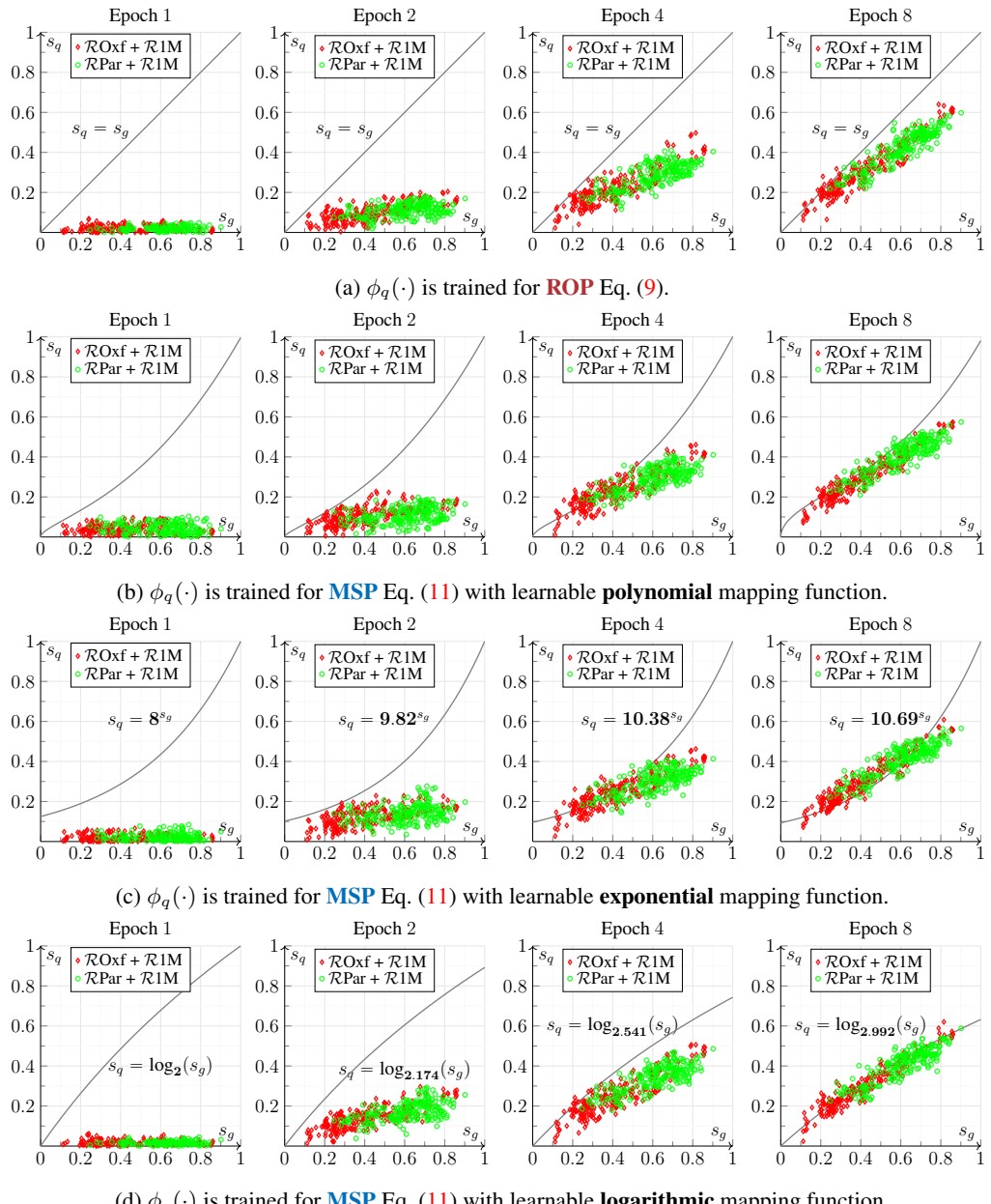

(a) $\phi_q(\cdot)$ is trained for **ROP** Eq. (9).

(b) $\phi_q(\cdot)$ is trained for **MSP** Eq. (11) with learnable **polynomial** mapping function.

(c) $\phi_q(\cdot)$ is trained for **MSP** Eq. (11) with learnable **exponential** mapping function.

(d) $\phi_q(\cdot)$ is trained for **MSP** Eq. (11) with learnable **logarithmic** mapping function.

Figure 8: **Visualization of the similarity score distributions.** SfM-120k, MobileNetV2 and GeM are adopted as the training dataset, query and gallery models, respectively. $s_g$: symmetric similarity score; $s_q$: asymmetric similarity score.

