# OpenReview forum: "A General Rank Preserving Framework for Asymmetric Image Retrieval"
_ICLR.cc/2023/Conference — ICLR 2023 poster_

### Official Review · Reviewer_BuFD · 2022-10-24

**Confidence:** 3
**Correctness:** 4
**Technical Novelty And Significance:** 3
**Empirical Novelty And Significance:** 3
**Recommendation:** 6

**Clarity, Quality, Novelty And Reproducibility:**

The presentation of this work is clear and easy to read. This work builds upon CSD (Wu et al., 2022b), however, the authors make the novelty of their work very clear in Page 6, and show that CSD is a special case of their method.

**Strength And Weaknesses:**

Strenghts:
- The paper is easy to read.
- The proposed method is well motivated, and makes sense for this problem.
- The authors justify their claims with thorough experiments and analysis.

Weaknesses:
- There are grammatical mistakes/typos in the paper that should be fixed, e.g. "we aims" in Page 3, "the top-K images are token
from F" (should be taken) etc.

**Summary Of The Paper:**

This paper is about asymmetric image retrieval, where the query features and the gallery features are extracted with different feature extractors. Gallery features typically use a more expensive feature extractor, whereas the query features are extracted with a more lightweight feature extractor to increase the speed during inference. The authors first compute the ranked list using the gallery features, and  add a consistency loss term such that the same ordering will be preserved when using the query features. Two solutions are proposed for preserving the ranked list ordering between different sets of features. The experiments are thorough and show that the proposed method brings improvements.

**Summary Of The Review:**

This paper proposes a novel method for preserving the ranked list for asymmetric image retrieval. The paper is easy to read, the proposed method looks correct, and the experiments confirm the claims made in the paper.

---

> ### Author Response · Authors · 2022-11-10
> **To reviewer BuFD**
>
> We would like to thank reviewer for an overall positive evaluation and general appreciation of our work. We correct typos and update the revised version. Welcome to continue the discussion with us.

---

### Official Review · Reviewer_vzrF · 2022-10-25

**Confidence:** 4
**Correctness:** 4
**Technical Novelty And Significance:** 2
**Empirical Novelty And Significance:** 3
**Recommendation:** 8

**Clarity, Quality, Novelty And Reproducibility:**

- The paper is well written and easy to understand.

- This work is reproducible.

- In real world application, the gallery model is continuously improved and their ranking for the same query will evolve overtime. I'm curious about the impact of that on this proposed framework and on asymmetric image retrieval methods in general.


**Strength And Weaknesses:**

Strengths of this paper:
- The proposed framework is more flexible than existing methods for asymmetric image retrieval, and doesn't require the query model to mimic the feature embeddings or the overall neighbor structures of the gallery model.
- This framework utilizes no annotation or labels during the training process, instead it directly optimizes the consistency of rank order between the query model and the gallery model.
- The paper proposes a comprehensive evaluation on 4 public datasets and the proposed framework showed to outperform state of the art methods by a significant margin.

Areas where the paper could improve on:
- The query model can only be as good as the gallery model is since no labels are used (depends on the quality of the gallery model).


**Summary Of The Paper:**

This paper proposes a generic rank preserving framework for asymmetric image retrieval. The rank preserving framework allows feature compatibility and order consistency between a query model and a gallery model simultaneously. Two methods are proposed as part of the rank preserving framework: i) the first realizes rank order preservation by directly preserving the consistency of the sorting results using a sigmoid function approximation, and ii) the second learns a monotonic mapping function between the returned similarity scores of query and gallery models. An evaluation of the proposed framework on 4 public datasets showed that that this framework outperforms significantly some state of the art methods.


**Summary Of The Review:**

This paper proposed a new and generic rank preserving framework for asymmetric image retrieval. The proposed framework doesn't require a labeled dataset and optimizes directly the consistency of rank order between the query model and the gallery model. Experimental results on 4 public datasets showed that the proposed framework outperforms state of the art methods. Therefore, the proposed framework looks solid and is novel enough.

---

> ### Author Response · Authors · 2022-11-10
> **To reviewer vzrF**
>
> We would like to thank the review’s constructive comments, positive rating and high score. We respond to the comments below and update the manuscript accordingly.
>
> **Q$_1$ : The query model can only be as good as the gallery model is since no labels are used (depends on the quality of the gallery model).**
>
> A$_1$ : Thanks for pointing out the limitation of our methods. In our framework, the gallery model is kept frozen without being optimized simultaneously when training the lightweight query model. As a result, the performance of the query model is heavily dependent on that of the gallery model. In the future, we will explore how to optimize both gallery and query models to achieve better retrieval accuracy and efficiency.
>
> **Q$_2$ : In real world application, the gallery model is continuously improved and their ranking for the same query will evolve overtime. I’m curious about the impact of that on this proposed framework and on asymmetric image retrieval methods in general.**
>
> A$_2$ : This is an interesting point. Currently, our framework does not consider that the gallery model will evolve over time. When the gallery model changes, we need to adjust the query model to make it compatible with the gallery model. In fact, several efforts (Shen et al., 2020 [1]; Zhang et al., 2022 [2]; Ramanujan et al., 2022 [3]; Wu et al., 2022 [4]) have been devoted to avoiding the painful "backfilling" process during model update. In the future, we will explore how to update the lightweight query model and the gallery model simultaneously.
>
> >[1] Yantao Shen, Yuanjun Xiong, Wei Xia, and Stefano Soatto. Towards backward-compatible representation learning. In *Proceedings of the IEEE Conference on Computer Vision and Pattern Recognition (CVPR)*, June 2020.
>
> >[2] Binjie Zhang, Yixiao Ge, Yantao Shen, Shupeng Su, FanziWu, Chun Yuan, Xuyuan Xu, YexinWang, and Ying Shan. Towards universal backward-compatible representation learning. In *Proceedings of the International Joint Conference on Artificial Intelligence, IJCAI*, pp. 1615–1621, 2022.
>
> >[3] Vivek Ramanujan, Pavan Kumar Anasosalu Vasu, Ali Farhadi, Oncel Tuzel, and Hadi Pouransari. Forward compatible training for large-scale embedding retrieval systems. In *Proceedings of the IEEE Conference on Computer Vision and Pattern Recognition (CVPR)*, pp. 19386–19395, 2022.
>
> >[4] Shengsen Wu, Liang Chen, Yihang Lou, Yan Bai, Tao Bai, Minghua Deng, and Ling-Yu Duan. Neighborhood consensus contrastive learning for backward-compatible representation. In *Proceedings of the AAAI Conference on Artificial Intelligence (AAAI)*, pp. 2722–2730, 2022.

---

### Official Review · Reviewer_BchT · 2022-10-26

**Confidence:** 4
**Correctness:** 4
**Technical Novelty And Significance:** 3
**Empirical Novelty And Significance:** 3
**Recommendation:** 8

**Clarity, Quality, Novelty And Reproducibility:**

As I mentioned above, the authors present their contributions in a very clear manner and the paper is very easy to follow. The method is also explained very clearly and it should be relativerly easy to reproduce, experiments incldued. Regarding the paper's contribution, I also think that their approach to solving this problem is novel.

**Strength And Weaknesses:**

The paper is well written and the method very well motivated by presenting a possible limitation of current methods (enforcing consistency of feature spaces) and proposing a direct solution to this problem by optimizing the consistency of the rank order instead, which by itself I think it is a very good idea. Then, figures and the explanation of their framework and methods are also very clear. Finally, experiments are extensive and includes a very useful ablation analysis.

There is however something that I would have liked to see in the paper, which is a deeper insight of the differences between the two methods for rank preserving proposed. For example, both methods are on par when using Sfm120K as training set but when using GLv2, MSP significantly outperforms ROP: why does this happen? Is there any scenario where one would choose ROP over MSP? It would have been interesting to see a deeper comparison between both methods.

A couple of typos I spotted:
- Page 3, last paragraph: we aims to learn -> we aim to learn
- Page 4, first paragraph: images are token -> images are taken


**Summary Of The Paper:**

The method addresses the problem of asymmetric image retrieval, in which different models are deployed for query and gallery - usally for computation efficiency reasons - and that consists on aligning their embedding spaces. The authors argue that current approaches, which try to enforce the consitency of features between models, are too limiting since this is too strict for lightweight query models with low capacity. To address this issue, instead of forcing the query model to produce the same representations as the gallery, the authors relax this problem and propose to optimize instead the consitency of the rank order. For this, they propose two methods that aim to achieve this rank consistency between two rank orders.

**Summary Of The Review:**

Even though I think that a deeper analysis comparing the methods proposed would have made the paper more well-rounded, it is already a very strong paper with a very strong contribution presented with a very clear motivation. Other than this weakness just mentioned, the experimental analysis is also quite extensive. Taking all this into account, I would recommend this for its acceptance.

---

> ### Author Response · Authors · 2022-11-10
> **To reviewer BchT**
>
> We thank the reviewer for the detailed and constructive comments. Below are our responses to all the questions and comments, and we have updated the manuscript accordingly. We hope that the responses are reasonable and satisfactory enough to address the reviewer’s concerns and we welcome further discussion.
>
> **Q$_1$ : Which is a deeper insight of the differences between the two methods for rank preserving proposed. For example, both methods are on par when using SfM-120K as training set but when using GLDv2, MSP significantly outperforms ROP: why does this happen?**
>
> A$_1$ : This is a good point. Besides the different objective functions of two instantiation methods, there are three extra factors including query model, gallery model and training dataset, that affect the final performance. In Figure 5 of the main paper, we show the impact of the gallery model and the query model, respectively. As for training dataset, its influence is shown in Tables 4, 6, 8 and 9 of the main paper. Here, we further enrich the experiments in the main paper to explore what exactly causes the difference in performance between two instantiation methods MSP and ROP. As shown in Table below, MSP achieves better results in most cases when GLDv2 is taken as the training set.
>
>  We believe that this should be related to the objective functions of the optimization. Although both MSP and ROP expect to keep the ranking order, ROP applies more strict restrictions. According to the definition of the indicator matrix in the main paper, ROP restricts the relative order between any two candidates in the ranking list. Here, we consider a situation when there are two candidates both similar to query images. Exchange of their ranking order does not cause a change to the current evaluation metric mAP. ROP strictly restricts the relative ranking order between any two candidates in the ranking list, which increases the learning difficulty of the lightweight query model. As for MSP, it pre-defines the target $\textbf{M}_g$, which is obtained by mapping $\textbf{S}_g$ with a learnable monotonic function, whose coefficients are updated during training. The query model only needs to align the corresponding asymmetric similarity $\textbf{S}_q$ to $\textbf{M}_g$, which does not explicitly consider the relative order between two candidates. GLDv2 has more diverse landmark images than SfM-120k. When it is taken as the training set, the learnable monotonic function captures a better distribution of similarity scores, which makes optimization of lightweight models much easier.
>
>  | Query&nbsp;Net | Training&nbsp;set| &nbsp;Method | GLDv2&nbsp;-&nbsp;Test |ROxf&nbsp;+&nbsp;R1M&ensp;(Medium) | ROxf&nbsp;+&nbsp;R1M&ensp;(Hard) |RPar&nbsp;+&nbsp;R1M&ensp;(Medium) | RPar&nbsp;+&nbsp;R1M&ensp;(Hard) |
>   |:------|:---------------:|:----------------:|:-------------:|:------------:|:---------------:|:------------------:|:----------------------------:|
>   | ShuffleNetv2 | GLDv2| **ROP**| **25.47** | 62.32 | 41.07 | 67.24 | **46.47** |
>   | ShuffleNetv2 | GLDv2| **MSP**| 25.53|**64.41** | **42.33** | **67.41** | 45.56 |
>   | MobiliNetV2 | GLDv2| **ROP**| **28.68**| 66.49 | 46.26 | 72.89 | 53.11 |
>   | MobiliNetV2 | GLDv2| **MSP**| 28.57|**68.68** | **47.04** | **75.16** | **55.97** |
>   | EfficientNetB3| GLDv2| **ROP**| 29.04|69.41 | 47.44 | 74.00 | 54.35 |
>   | EfficientNetB3| GLDv2| **MSP**| **29.75**| **70.98** | **50.64** | **76.63** | **56.92** |
>
> **Q$_2$ : Is there any scenario where one would choose ROP over MSP?**
>
> A$_2$ : This is an interesting question. According to the analysis above, ROP constrains more strict
> restriction than MSP. Currently, the testing datasets used in the main paper treat different positive
> samples equally, while in fact there is still a relative ranking order between the similarities of different
> positive candidates to the query. If there is a scenario where we need to strictly distinguish the
> ranking order between positive and query samples, the restriction of ROP may be a better choice.

---

> > ### Comment · Reviewer_BchT · 2022-11-18
> > **To the authors**
> >
> > I would like to sincerely thank the authors for this very thorough response to my concerns - I really appreciate the effort on running these extra experiments. Their response satisfactorily answer my question regarding the practical differences between ROP and MSP.

---

> > > ### Author Response · Authors · 2022-11-19
> > > **Thanks!**
> > >
> > > Thanks again for your efforts on improving our work.

---

### Official Review · Reviewer_6hUg · 2022-10-27

**Confidence:** 3
**Clarity, Quality, Novelty And Reproducibility:** The paper is well-written, and the pr…
**Correctness:** 3
**Technical Novelty And Significance:** 3
**Empirical Novelty And Significance:** 3
**Recommendation:** 6

**Strength And Weaknesses:**

Pros:
The empirical findings support the efficacy of the proposed solution. The ablation study and the extra experiments depicted in the appendix aid in understanding the core concepts of the algorithm.
The paper is well-written, and the premise is original and novel.

Minor remarks:
The metric used for ranking the gallery images is not denoted.
Proof that the similarity scores of the gallery satisfy the monotonically decreasing property could be useful.
Minor typos, especially in section 4.

**Summary Of The Paper:**

This paper presents a method for asymmetric image retrieval. The main goal is to utilize a lightweight CNN without affecting the retrieval performance. A sizeable pretrained model is employed to extract features from the dataset, and a ranking list is assembled by retrieving the most similar images to the query image. Then, the asymmetric similarity scores are calculated using the features of the images from the ranking list and the features of the query image from the lightweight model. Finally, to ensure that the original ranking is preserved, two different techniques are proposed: Rank Order Preservation and Monotonic Similarity Preservation. The first technique optimizes the consistency of rank, while the second uses a learnable function to map the similarity scores of both models.

**Summary Of The Review:**

Pros:
The empirical findings support the efficacy of the proposed solution. The ablation study and the extra experiments depicted in the appendix aid in understanding the core concepts of the algorithm.
The paper is well-written, and the premise is original and novel.

Minor remarks:
The metric used for ranking the gallery images is not denoted.
Proof that the similarity scores of the gallery satisfy the monotonically decreasing property could be useful.
Minor typos, especially in section 4.

---

> ### Author Response · Authors · 2022-11-10
> **To reviewer 6hUg**
>
> We thank the reviewer for the constructive feedback and detailed review. We also thank the reviewer for pointing out some typos, which helps us to improve the writing of Section 4. We answer all questions below and revise the paper accordingly.
>
> **Q$_1$: The metric used for ranking the gallery images is not denoted.**
>
> A$_1$ : In this paper, we assume that both the query and gallery models map images into L2-normalized features. The **cosine similarities** between query and gallery features are calculated as ranking metric to measure the similarities between query and gallery images. A large cosine similarity means that two images are similar and vice versa.
>
> **Q$_2$ : Proof that the similarity scores of the gallery satisfy the monotonically decreasing property could be useful.**
>
> A$_2$ : Image retrieval aims to sort gallery images based on their similarities to the given query image. It typically consists of two steps including calculating the similarities and ranking the gallery images with the corresponding similarities in decreasing order. Since the similarity scores $\textbf{S}_g$ of the gallery images in the main paper is obtained by symmetric retrieval, it has been sorted in decreasing order. As for the asymmetric scores $\textbf{S}_q$, it has no consistent decreasing guarantee. However, our framework restricts asymmetric retrieval to preserve the ranking order of symmetric retrieval, which to some extent maintains the monotonicity of the original ranking order, *i.e.*, monotonically decreasing.

---

### Decision · Program_Chairs · 2023-01-20

**Decision:**

Accept: poster

**Justification For Why Not Higher Score:**

- Although interesting, asymmetric image retrieval can be regarded as a niche
- The sigmoid is used to approximate the Heaviside function in smooth rank approximation, although methods with more theoretical guarantees, (Ramzi et al., 2021)
- Fixed gallery model

**Justification For Why Not Lower Score:**

- Sound methods (ROP and MSP) for smooth ranking consistency
- Thorough analysis of the preferred ROP vs MSP method
- Convincing experimental validation

**Metareview: Summary, Strengths And Weaknesses:**

This paper deals with asymmetric image retrieval, which goal is to align the off-line computed embeddings of a large gallery dataset with the embedding of the query computed online with a lightweight model. In contrast to previous work, which directly align the similarities between the gallery and query models, the authors propose a more flexible and robust approach consisting in preserving the rank between them. Two solutions are proposed for enforcing the ranking consistency: Rank Order Preservation (ROP), which uses a smooth rank approximation, and Monotonic Similarity Preservation (MSP) which maps the similarities of query and gallery models. Experiments are conducted on four public datasets.
The paper initially received positive reviews, with two accept (8) recommendations, and two borderline accept (6) recommendations. The reviewers appreciated the originality of the approach and the good experimental results. Reviewers requested to deepen the difference between ROP and MSP, and to discuss the impact of the approach with a variable gallery model. The rebuttal did a good job in answering to these concerns, and there was a consensus to accept the paper after the discussion period.

The AC carefully reads the submission and discussions. The AC considers that the contributions of the paper are solid and well justified. The proposed MSP approach provides an interesting generalization of the recent CSD approach to the non-linear case. The discussion to the differences between ROP and MSP provided in the rebuttal brings further clarifications on which methods can be preferred depending on the context. The experiments are convincing and draws several interesting perspectives for future works in asymmetric image retrieval. Therefore, the AC recommends acceptance.

**Note From Pc:**

if the above contains the word "oral" or "spotlight" please see: "oral" presentation means -> notable-top-5% and "spotlight" means -> notable-top-25%. As stated in our emails, we are disassociating presentation type from AC recommendations